# PROVABLE REPRESENTATION LEARNING FOR IMITATION LEARNING VIA BI-LEVEL OPTIMIZATION

## ABSTRACT

A common strategy in modern learning systems is to learn a representation which is useful for many tasks, a.k.a. representation learning. We study this strategy in the imitation learning setting for Markov decision processes (MDPs) where multiple experts' trajectories are available. We formulate representation learning as a bi-level optimization problem where the "outer" optimization tries to learn the joint representation and the "inner" optimization encodes the imitation learning setup and tries to learn task-specific parameters. We instantiate this framework for the imitation learning settings of behavior cloning and observation-alone. Theoretically, we provably show using our framework that representation learning can reduce the sample complexity of imitation learning in both settings. We also provide proof-of-concept experiments to verify our theoretical findings.

## 1 INTRODUCTION

Humans can often learn from experts quickly and with a few demonstrations and we would like our artificial agents to do the same. However, even for simple imitation learning tasks, the current state-of-the-art methods require thousand of demonstrations. Humans do not learn new skills from scratch. We can summarize learned skills, distill them and build a common ground, a.k.a, representation that is useful for learning future skills. Can we build an agent to do the same?

The current paper studies how to apply representation learning to imitation learning. Specifically, we want to build an agent that is able learn a representation from multiple experts' demonstrations, where the experts aim to solve different Markov decision processes (MDPs) that share the same state and action spaces but can differ in the transition and reward functions. The agent can use this representation to reduce the number of demonstrations required for a new imitation learning task. While several methods have been proposed (Duan et al., 2017; Finn et al., 2017b; James et al., 2018) to build agents that can adapt quickly to new tasks, none of them, to our knowledge, give provable guarantees showing the benefit of using past experience. Furthermore, they do not focus on learning a representation. See Section 2 for more discussions.

In this paper, we propose a framework to formulate this problem and analyze the statistical gains of representation learning. The main idea is to use bi-level optimization formulation where the "outer" optimization tries to learn the joint representation and the "inner" optimization encodes the imitation learning setup and tries to learn task-specific parameters. In particular, the inner optimization is flexible enough to allow the agent to interact with the environment. This framework allows us to do a rigorous analysis to show provable benefits of representation learning for imitation learning. With this framework at hand, we make the following concrete contributions:

- We first instantiate our framework in the setting where the agent can observe experts' actions and tries to find a policy that matches the expert's policy, a.k.a, behavior cloning. This setting can be viewed as a straightforward extension of multi-task representation learning for supervised learning (Maurer et al., 2016). We show in this setting that with sufficient number of experts (possibly optimizing for different reward functions), the agent can learn a representation that provably reduces the sample complexity for a new target imitation learning task.
- Next, we consider a more challenging setting where the agent *cannot* observe experts' actions but only their states, a.k.a., the observation-alone setting. We set the inner optimization as a min-max problem inspired by Sun et al. (2019). Notably, this min-max problem requires the agent to interact with the environment to collect samples. We again show that with sufficient number of

experts, the agent can learn a representation that provably reduces the sample complexity for a target task where the agent cannot observe actions from either source experts or the target expert.

- We conduct experiments to verify our theoretical insights by learning a representation from multiple tasks using our framework and testing it using both behavior cloning and policy optimization. In these settings, we observe that by learning representations the agent can learn a good policy with fewer samples than needed to learn a policy from scratch.

The key contribution is to connect existing literature on multi-task representation learning that deals with supervised learning (Maurer et al., 2016) to single task imitation learning methods with guarantees (Syed & Schapire, 2010; Ross et al., 2011; Sun et al., 2019). To our knowledge, this is the first work showing such guarantees for general losses that are not necessarily convex.

## 2    RELATED WORK

Representation learning has shown its great power in various domains. See Bengio et al. (2013) for a survey. Theoretically, Maurer et al. (2016) gave analysis showing representation can provably reduce the sample complexity in the multi-task supervised learning setting. Recently, Arora et al. (2019) analyzed the benefit of representation learning via contrastive learning. These papers all build representations for the agent / learner. We remark that researchers also try to build representations about the environment / physical world (Wu et al., 2017).

Imitation learning can help with sample efficiency of many problems (Ross & Bagnell, 2010; Sun et al., 2017; Daumé et al., 2009; Chang et al., 2015; Pan et al., 2018). Most existing work consider the setting where the learner can observe expert's action. A general strategy is use supervised learning to learn a policy that maps the state to action that matches expert's behaviors. The most straightforward one is behavior cloning (Pomerleau, 1991), which we also study in our paper. More advanced approaches have also been proposed (Ross et al., 2011; Ross & Bagnell, 2014; Sun et al., 2018). These approaches, including behavior cloning, often enjoy sound theoretical guarantees in the single task case. Our paper extends the theoretical guarantees of behavior cloning to the multi-task representation learning setting.

This paper also considers a more challenging setting, imitation learning from observation alone. Though some model-based methods have been proposed (Torabi et al., 2018; Edwards et al., 2018), these methods lack theoretical guarantees. Another line of work learns a policy that minimizes the difference between the state distributions induced by it and the expert policy, under a certain distributional metric (Ho & Ermon, 2016). Sun et al. (2019) gave a theoretical analysis to characterize the sample complexity of this approach and our method for this setting is inspired by their approach.

A line of work uses meta-learning for imitation learning (Duan et al., 2017; Finn et al., 2017b; James et al., 2018). Our work is different from theirs as we want to explicitly learn a representation that is useful across all tasks whereas these work try to learn a meta-algorithm that can quickly adapt to a new task. For example, Finn et al. (2017b) used a gradient based method for adaptation. Recently Raghu et al. (2019) argued that most of the power of MAML (Finn et al., 2017a) like approaches comes from learning a shared representation.

On the theoretical side of meta-learning and multi-task learning, Baxter (2000) performed the first theoretical analysis and gave sample complexity bounds using covering numbers. Bullins et al. (2019) provides an efficient algorithm that generalizes to new unseen tasks, but for linear representations. Another recent line of work analyzes gradient based meta-learning methods, similar to MAML (Finn et al., 2017a). Existing work on the sample complexity and regret of these methods (Denevi et al., 2019; Finn et al., 2019; Khodak et al., 2019) show guarantees for convex losses by leveraging tools from online convex optimization. In contrast, our analysis works for arbitrary function classes and the bounds depend on the gaussian averages of these classes. Recent work (Rajeswaran et al., 2019) uses a bi-level optimization framework for meta-learning and improves computation (not statistical) aspects of meta-learning through implicit differentiation.

## 3    PRELIMINARIES

**Markov Decision Processes (MDPs):**    Let $\mathcal{M} = (\mathcal{S}, \mathcal{A}, P, C, \nu)$ be an MDP, where $\mathcal{S}$ is the state space, $\mathcal{A}$ is the finite action space with $|\mathcal{A}| = K$, $H \in \mathbb{Z}_+$ is the planning horizon, $P : \mathcal{S} \times \mathcal{A} \rightarrow$

$\triangle(\mathcal{S})$ is the transition function, $C : \mathcal{S} \times \mathcal{A} \to \mathbb{R}$ is the cost function and $\nu \in \triangle(S)$ is the initial state distribution. We assume that cost is bounded by 1, i.e. $C(s,a) \leq 1, \forall s \in \mathcal{S}, a \in \mathcal{A}$. This is a standard regularity condition used in many theoretical reinforcement learning work. A (stochastic) policy is defined as $\boldsymbol{\pi} = (\pi_1, \ldots, \pi_H)$, where $\pi_h : \mathcal{S} \to \triangle(\mathcal{A})$ prescribes a distribution over action for each state at level $h \in [H]$. For a stationary policy, we have $\pi_1 = \cdots = \pi_H = \pi$. A policy $\boldsymbol{\pi}$ induces a random trajectory $s_1, a_1, s_2, a_2, \ldots, s_H, a_H$ where $s_1 \sim \nu, a_1 \sim \pi_1(s), s_2 \sim P_{s_1,a_1}$ etc. Let $\nu_h^{\boldsymbol{\pi}}$ denote the distribution over $\mathcal{S}$ induced at level $h$ by policy $\boldsymbol{\pi}$. The value function $V_h^{\boldsymbol{\pi}} : \mathcal{S} \to \mathbb{R}$ is defined as

$$V_h^{\boldsymbol{\pi}}(s_h) = \mathbb{E}\left[\sum_{i=h}^{H} C(s_i, a_i) \mid a_i \sim \pi_i(s_i), s_{i+1} \sim P_{s_i,a_i}\right]$$

and the state-action function $Q_h^{\boldsymbol{\pi}}(s_h, a_h)$ is defined as $Q_h^{\boldsymbol{\pi}}(s_h, a_h) = \mathbb{E}_{s_{h+1} \sim P_{s_h,a_h}}[V_h^{\boldsymbol{\pi}}(s_{h+1})]$. The goal is to learn a policy $\boldsymbol{\pi}$ that minimizes the expected cost $J(\boldsymbol{\pi}) = \mathbb{E}_{s_1 \sim \nu} V_1^{\boldsymbol{\pi}}(s_1)$. We define the Bellman operator at level $h$ for any policy $\boldsymbol{\pi}$ as $\Gamma_h^{\boldsymbol{\pi}} : \mathbb{R}^{\mathcal{S}} \to \mathbb{R}^{\mathcal{S}}$, where for $s \in \mathcal{S}$ and $g \in \mathbb{R}^{\mathcal{S}}$,

$$(\Gamma_h^{\boldsymbol{\pi}} g)(s) \coloneqq \mathbb{E}_{a \sim \pi_h(s), s' \sim P_{s,a}}[g(s')] \tag{1}$$

**Multi-task Imitation learning:** We formally describe the problem we want to study. We assume there are multiple tasks (MDPs) sampled i.i.d. from a distribution $\eta$. A task $\mu \sim \eta$ is an MDP $\mathcal{M}_\mu = (\mathcal{S}, \mathcal{A}, H, P_\mu, C_\mu, \nu_\mu)$; all tasks share everything except the cost function, initial state distribution and transition function. For simplicity of presentation, we will assume a common transition function $P$ for all tasks; proofs remain exactly the same even otherwise. For every task $\mu$, $\boldsymbol{\pi}_\mu^* = (\pi_{1,\mu}^*, \ldots, \pi_{H,\mu}^*)$ is an *expert policy* that the learner has access to in the form of trajectories induced by that policy. The trajectories may or may not contain expert's actions. These correspond to two settings that we discuss in more detail in Section 5 and Section 6. The distributions of states induced by this policy at different levels are denoted by $\{\nu_{1,\mu}^*, \ldots, \nu_{H,\mu}^*\}$ and the average state distribution as $\nu_\mu^* = \frac{1}{H}\sum_{h=1}^{H} \nu_{h,\mu}^*$. We define $V_{h,\mu}^*$ to be the value function of $\boldsymbol{\pi}_\mu^*$ and $J_\mu$ to be the expected cost function for task $\mu$. We will drop the subscript $\mu$ whenever the task at hand is clear from context. Of interest is also the special case where the expert policy $\boldsymbol{\pi}_\mu^*$ is stationary.

**Representation learning:** In this work, we wish to learn policies from a function class of the form $\Pi = \mathcal{F} \circ \Phi$, where $\Phi \subseteq \{\phi : \mathcal{S} \to \mathbb{R}^d \mid \|\phi(s)\|_2 \leq R\}$ is a class of bounded norm *representation functions* mapping states to vectors and $\mathcal{F} \subseteq \{f : \mathbb{R}^d \to \triangle(\mathcal{A})\}$ is a class of functions mapping state representations to distribution over actions. We will be using linear functions, i.e. $\mathcal{F} = \{x \to \texttt{softmax}(Wx) \mid W \in \mathbb{R}^{K \times d}, \|W\|_F \leq 1\}$. We denote a policy parametrized by $\phi \in \Phi$ and $f \in \mathcal{F}$ by $\pi^{\phi,f}$, where $\pi^{\phi,f}(a|s) = f(\phi(s))_a$. In some cases, we may also use the policy $\pi^{\phi,f}(a|s) = \mathbb{I}\{a = \arg\max_{a' \in A} f(\phi(s))_{a'}\}$[1]. Denote $\Pi^\phi = \{\pi^{\phi,f} : f \in \mathcal{F}\}$ to be the class of policies that use $\phi$ as the representation function.

Given demonstrations from expert policies for $T$ tasks sampled independently from $\eta$, we wish to first learn representation functions $(\hat{\phi}_1, \ldots, \hat{\phi}_H)$ so that we can use a few demonstrations from an expert policy $\boldsymbol{\pi}^*$ for new task $\mu \sim \eta$ and learn a policy $\boldsymbol{\pi} = (\pi_1, \ldots, \pi_H)$ that uses the learned representations, i.e. $\pi_h \in \Pi^{\hat{\phi}_h}$, such that has average cost of $\boldsymbol{\pi}$ is not too far away from $\boldsymbol{\pi}^*$. In the case of stationary policies, we need to learn a single $\phi$ by using tasks and learn $\pi \in \Pi^\phi$ for a new task. The hope is that data from multiple tasks can be used to learn a complicated function $\phi \in \Phi$ first, thus requiring only a few samples for a new task to learn a linear policy from the class $\Pi^\phi$.

**Gaussian complexity:** As in Maurer et al. (2016), we measure the complexity of a function class $\mathcal{H} \subseteq \{h : \mathcal{X} \to \mathbb{R}^d\}$ on a set $\mathbf{X} = (X_1, \ldots, X_n) \in \mathcal{X}^n$ by using the following Gaussian average

$$G(\mathcal{H}(\mathbf{X})) = \mathbb{E}\left[\sup_{h \in \mathcal{H}} \sum_{\substack{i=1 \\ j=1}}^{d,n} \gamma_{ij} h_i(X_j) \mid X_j\right] \tag{2}$$

where $\gamma_{ij}$ are independent standard normal variables. Bartlett & Mendelson (2003) also used Gaussian averages to show some generalization bounds.

---

[1] Break ties in any way

## 4 BI-LEVEL OPTIMIZATION FRAMEWORK

In this section we introduce our framework and give a high-level description of the conditions under which this framework gives us statistical guarantees. Our main idea is to phrase learning representations for imitation learning as the following bi-level optimization

$$\min_{\phi \in \Phi} L(\phi) := \mathop{\mathbb{E}}_{\mu \sim \eta} \min_{\pi \in \Pi^\phi} \ell^\mu(\pi) \tag{3}$$

Here $\ell^\mu$ is the *inner* loss function that penalizes $\pi$ being different from $\pi_\mu^*$ for the task $\mu$. In general, one can use any loss $\ell^\mu$ that is used for single task imitation learning, e.g. for the behavioral cloning setting (cf. Section 5), $\ell^\mu$ is a classification like loss that penalizes the mismatch between predictions by $\pi^*$ and $\pi$, while for the observation-alone setting (cf. Section 6) it is some measure of distance between the state visitation distributions induced by $\pi$ and $\pi^*$. The *outer* loss function is over the representation $\phi$. The use of bi-level optimization framework naturally enforces policies in the inner optimization to share the same representation.

While Equation 3 is formulated in terms of the distribution $\eta$, in practice we only have access to few samples for $T$ tasks; let $\mathbf{x}^{(1)}, \ldots, \mathbf{x}^{(T)}$ denote samples from tasks $\mu^{(1)}, \ldots, \mu^{(T)}$ sampled i.i.d. from $\eta$. We thus learn the representation $\hat{\phi}$ by minimizing empirical version $\hat{L}$ of Equation 3.

$$\hat{L}(\phi) = \frac{1}{T} \sum_{i=1}^T \min_{\pi \in \Pi^\phi} \ell^{\mathbf{x}^{(i)}}(\pi) = \frac{1}{T} \sum_{i=1}^T \ell^{\mathbf{x}^{(i)}}(\pi^{\phi, \mathbf{x}^{(i)}})$$

where $\ell^{\mathbf{x}}$ is the empirical loss on samples $\mathbf{x}$ and $\pi^{\phi, \mathbf{x}} = \arg\min_{\pi \in \Pi^\phi} \ell^{\mathbf{x}}(\pi)$ corresponds to a task specific policy that uses a fixed representation $\phi$. Our goal then is to show that for a new task $\mu \sim \eta$, the policy $\pi^{\hat{\phi}, \mathbf{x}}$ learned by using samples $\mathbf{x}$ from the task $\mu$ has low expected cost $J_\mu$, i.e.,

**Informal Theorem 4.1.** *With high probability over the sampling of train task data and with sufficient number of tasks and samples per task,*

$$\mathop{\mathbb{E}}_{\mu \sim \eta} \mathop{\mathbb{E}}_{\mathbf{x}} J_\mu(\pi^{\hat{\phi}, \mathbf{x}}) - \mathop{\mathbb{E}}_{\mu \sim \eta} J_\mu(\pi_\mu^*) \text{ is small}$$

At a high level, in order to prove such a theorem for a particular choice of $\ell^\mu$, we would need to prove the following three properties about $\ell^\mu$ and $\ell^{\mathbf{x}}$:

1. $\ell^{\mathbf{x}}(\pi)$ concentrates to $\ell^\mu(\pi)$ simultaneously for all $\pi \in \Pi^\phi$ (for a fixed $\phi$), with sample complexity depending on some complexity measure of $\Pi^\phi$ rather than being polynomial in $|\mathcal{S}|$;
2. a small value of $\ell^\mu(\pi)$ implies a small value for $J_\mu(\pi) - J_\mu(\pi_\mu^*)$;
3. if $\phi$ and $\phi'$ induce "similar" representations then $\min_{\pi \in \Pi^\phi} \ell^\mu(\pi)$ and $\min_{\pi \in \Pi^{\phi'}} \ell^\mu(\pi)$ are close.

The first property ensures that learning a policy for a single task by fixing the representation is sample efficient, thus making representation learning a useful problem to solve. The second property ensures that matching the behavior of the expert as measured by the loss $\ell^\mu$ ensures low average cost i.e., $\ell^\mu$ is meaningful for the average cost; any standard imitation learning loss will satisfy this. The third property is specific to representation learning and requires $\ell^\mu$ to use representations in a smooth way. This ensures that the empirical loss for $T$ tasks is a good estimate for the average loss on tasks sampled from $\eta$. We *prove* these three properties for the cases where $\ell^\mu$ is the either behavioral cloning loss or observation-alone loss, with natural choices for the empirical loss $\ell^{\mathbf{x}}$. However the general proof recipe can be used for potentially many other settings and loss functions.

In the next section, we will describe representation learning for behavioral cloning as an instantiation of the above framework and describe the various components of the framework. Furthermore we will describe the results and give a proof sketch to show how the aforementioned properties help us show our final guarantees. The guarantees for this setting follow almost directly from results in Maurer et al. (2016) and Ross et al. (2011). Later in Section 6 we describe the same for the observations alone setting which is more non-trivial.

## 5 REPRESENTATION LEARNING FOR BEHAVIORAL CLONING

**Choice of $\ell^\mu$:** We first specify the inner loss function in the bi-level optimization framework. In the single task setting, the goal of behavioral cloning (Syed & Schapire, 2010; Ross et al., 2011)

is to use expert trajectories of the form $\tau = (s_1, a_1, \ldots, s_H, a_H)$ to learn a stationary policy[2] that tries to mimic the decisions of the expert policy on the states visited by the expert. For a task $\mu$, this reduces to a supervised classification problem that minimizes a surrogate to the following loss $\ell_{0-1}^\mu(\pi) = \mathbb{E}_{s \sim \nu_\mu^*, a \sim \pi_\mu^*(s)} \mathbb{I}\{\pi(s) \neq a\}$. We abuse notation and denote this distribution over $(s, a)$ for task $\mu$ as $\mu$; so $(s, a) \sim \mu$ is the same as $s \sim \nu_\mu^*, a \sim \pi_\mu^*(s)$. Prior work (Syed & Schapire, 2010; Ross et al., 2011) have shown that a small value of $\ell_{0-1}^\mu(\pi)$ implies a small difference $J(\pi) - J(\pi^*)$. Thus for our setting, we choose $\ell^\mu$ to be of the following form

$$\ell^\mu(\pi) = \mathop{\mathbb{E}}_{s \sim \nu_\mu^*, a \sim \pi_\mu^*(s)} \ell(\pi(s), a) = \mathop{\mathbb{E}}_{(s,a) \sim \mu} \ell(\pi(s), a) \tag{4}$$

where $\ell$ is any surrogate to 0-1 loss $\mathbb{I}\{a \neq \arg\max_{a' \in A} \pi(s)_{a'}\}$ that is *Lipschitz* in $\phi(s)$. In this work we consider the logistic loss $\ell(\pi(s), a) = -\log(\pi(s)_a)$.

**Learning $\phi$ from samples:** Given expert trajectories for $T$ tasks $\mu^{(1)}, \ldots, \mu^{(T)}$ we construct a dataset $\mathbf{X} = \{\mathbf{x}^{(1)}, \ldots, \mathbf{x}^{(T)}\}$, where $\mathbf{x}^{(t)} = \{(s_j^t, a_j^t)\}_{j=1}^n \sim (\mu^{(t)})^n$ is the dataset for task $t$. Details of the dataset construction are provided in Section C.1. Let $\mathbf{S}$ denote the set of states $\{s_j^t\}$. Instantiating our framework, we learn a good representation by solving $\hat{\phi} = \arg\min_{\phi \in \Phi} \hat{L}(\phi)$, where

$$\hat{L}(\phi) := \frac{1}{T} \sum_{t=1}^T \min_{\pi \in \Pi^\phi} \frac{1}{n} \sum_{j=1}^n \ell(\pi(s_j^t), a_j^t) = \frac{1}{T} \sum_{t=1}^T \min_{\pi \in \Pi^\phi} \hat{\ell}^{\mathbf{x}^{(t)}}(\pi) \tag{5}$$

where $\ell^\mathbf{x}$ is loss on samples $\mathbf{x} = \{(s_j, a_j)\}_{j=1}^n$ defined as $\ell^\mathbf{x}(\pi) = \frac{1}{n} \sum_{j=1}^n \ell(\pi(s_j), a_j)$.

**Evaluating representation $\hat{\phi}$:** A learned representation $\hat{\phi}$ is tested on a new task $\mu \sim \eta$ as follows: draw samples $\mathbf{x} \sim \mu^n$ using trajectories from $\pi_\mu^*$ and solve $\pi^{\hat{\phi}, \mathbf{x}} = \arg\min_{\pi \in \Pi^{\hat{\phi}}} \hat{\ell}^\mathbf{x}(\pi)$. Does $\pi^{\hat{\phi}, \mathbf{x}}$ have expected cost $J_\mu(\pi^{\hat{\phi}, \mathbf{x}})$ not much larger than $J_\mu(\pi_\mu^*)$? The following theorem answers this question. We make the following two assumptions to prove the theorem.

**Assumption 5.1.** *The expert policy $\pi_\mu^*$ is deterministic for every $\mu \in support(\eta)$.*

**Assumption 5.2** (Policy realizability)**.** *There is a representation $\phi^* \in \Phi$ such that for every $\mu \in support(\eta)$, $\pi_\mu \in \Pi^{\phi^*}$ such that $\pi_\mu(s)_{\pi_\mu^*(s)}{}^3 \geq 1 - \gamma, \forall s \in \mathcal{S}$ for some $\gamma < 1/2$.*

The first assumption holds if $\pi_\mu^*$ is aiming to maximize some cost function. The second assumption is for representation learning to make sense: we need to assume the existence of a common representation $\phi^*$ that can approximate all expert policies and $\gamma$ measures this expressiveness of $\Phi$. Now we present our first main result.

**Theorem 5.1.** *Let $\hat{\phi} \in \arg\min_{\phi \in \Phi} \hat{L}(\phi)$. Under Assumptions 5.1, 5.2, with probability $1 - \delta$ over the sampling of dataset $\mathbf{X}$, we have*

$$\mathop{\mathbb{E}}_{\mu \sim \eta} \mathop{\mathbb{E}}_{\mathbf{x} \sim \mu^n} J_\mu(\pi^{\hat{\phi}, \mathbf{x}}) - \mathop{\mathbb{E}}_{\mu \sim \eta} J_\mu(\pi_\mu^*) \leq H^2(2\gamma + \epsilon_{gen})$$

*where $\epsilon_{gen} = c \frac{G(\Phi(\mathbf{S}))}{T\sqrt{n}} + c' \frac{R\sqrt{K}}{\sqrt{n}} + c'' \sqrt{\frac{\ln(4/\delta)}{T}}$, for some small constants $c, c', c''$.*

To gain intuition for what the above bound means, we give a PAC-style guarantee for the special case where the class of representation functions $\Phi$ is finite. This follows directly from the above theorem and the use of Massart's lemma.

**Corollary 5.1.** *In the same setting as Theorem 5.1, suppose $\Phi$ is finite. If number of tasks satisfies $T \geq c_1 \max \left\{ \frac{H^4 R^2 \log(|\Phi|)}{\epsilon^2}, \frac{H^4 \ln(4/\delta)}{\epsilon^2} \right\}$, and number of samples (expert trajectories) per task satisfies $n \geq c_2 \frac{H^4 R^2 K}{\epsilon^2}$ for small constants $c_1, c_2$, then with probability $1 - \delta$,*

$$\mathop{\mathbb{E}}_{\mu \sim \eta} \mathop{\mathbb{E}}_{\mathbf{x} \sim \mu^n} J_\mu(\pi^{\hat{\phi}, \mathbf{x}}) - \mathop{\mathbb{E}}_{\mu \sim \eta} J_\mu(\pi_\mu^*) \leq H^2 \gamma + \epsilon$$

---

[2] We can easily extend the theory to non-stationary policies

[3] We abuse notation and use $\pi_\mu^*(s)$ instead of $\arg\max_{a \in \mathcal{A}} \pi_\mu^*(s)_a$

**Discussion:** The above bound says that as long as we have enough tasks to learn a representation from $\Phi$ and sufficient samples per task to learn a linear policy, the learned policy will have small average cost on a new task from $\eta$. The first term $H^2\gamma$ is small if the representation class $\Phi$ is expressive enough to approximate the expert policies (see Assumption 5.2). The results says that if we have access to data from $T = O\left(\frac{H^4 R^2 \log(|\Phi|)}{\epsilon^2}\right)$ tasks sampled from $\eta$, we can use them to learn a representation such that for a new task we only need $n = O\left(\frac{H^4 R^2 K}{\epsilon^2}\right)$ samples (expert demonstrations) to learn a linear policy with good performance. In contrast, without access to tasks, we would need $n = O\left(\max\left\{\frac{H^4 R^2 \log(|\Phi|)}{\epsilon^2}, \frac{H^4 R^2 K}{\epsilon^2}\right\}\right)$ samples from the task to learn a good policy $\pi \in \Pi$ from scratch. Thus if the complexity of the representation function class $\Phi$ is much more than number of actions ($\log(|\Phi|) \gg K$ in this case), then multi-task representation learning might be much more sample efficient[4]. Note that the dependence of sample complexity on $H$ comes from the error propagation when going from $\ell^\mu$ to $J_\mu$; this is also observed in single task imitation learning (Ross et al., 2011; Sun et al., 2019).

We give a proof sketch for Theorem 5.1 below, while the full proof is deferred to Appendix A.

## 5.1 PROOF SKETCH

The proof has two main steps. In the first step we bound the error due to use of samples. The policy $\pi^{\phi,\mathbf{x}}$ that is learned on samples $\mathbf{x} \sim \mu^n$ is evaluated on the distribution $\mu$ and the average loss incurred by representation $\phi$ across tasks is $\bar{L}(\phi) = \underset{\mu \sim \eta}{\mathbb{E}} \underset{\mathbf{x} \sim \mu^n}{\mathbb{E}} \ell^\mu(\pi^{\phi,\mathbf{x}})$.

On the other hand, if the learner had complete access to the distribution $\eta$ and distributions $\mu$ for every task, then the loss minimizer would be $\phi^* = \arg\min_{\phi \in \Phi} L(\phi)$, where $L(\phi) \coloneqq \underset{\mu \sim \eta}{\mathbb{E}} \min_{\pi \in \Pi^\phi} \ell^\mu(\pi)$.

Using results from Maurer et al. (2016), we can prove the following about $\hat{\phi}$

**Lemma 5.2.** *With probability $1 - \delta$ over the choice of $\mathbf{X}$, $\hat{\phi} \in \arg\min_{\phi \in \Phi} \hat{L}(\phi)$ satisfies*

$$\bar{L}(\hat{\phi}) \leq \min_{\phi \in \Phi} L(\phi) + c\frac{G(\Phi(\{s_j^t\}))}{T\sqrt{n}} + c'\frac{R\sqrt{K}}{\sqrt{n}} + c''\sqrt{\frac{\ln(1/\delta)}{T}}$$

The proof of this lemma is provided in the appendix for completeness.

The second step of the proof is connecting the loss $\bar{L}(\phi)$ and the average cost $J_\mu$ of the policies induced by $\phi$ for tasks $\mu \sim \eta$. This can obtained by using the connection between the surrogate 0-1 loss $\ell^\mu$ and the cost $J_\mu$ that has been established in prior work (Ross et al., 2011; Syed & Schapire, 2010). The following lemma uses the result for deterministic expert policies from Ross et al. (2011).

**Lemma 5.3.** *Given a representation $\phi$ with $\bar{L}(\phi) \leq \epsilon$. Let $\mathbf{x} \sim \mu^n$ be samples for a new task $\mu \sim \eta$. Let $\pi^{\phi,\mathbf{x}}$ be the policy learned by behavioral cloning on the samples, then under Assumption 5.1*

$$\underset{\mu \sim \eta}{\mathbb{E}} \underset{\mathbf{x} \sim \mu^n}{\mathbb{E}} J_\mu(\pi^{\phi,\mathbf{x}}) - \underset{\mu \sim \eta}{\mathbb{E}} J_\mu(\pi_\mu^*) \leq H^2\epsilon$$

This suggests that making $\bar{L}$ small is good enough. A simple implication of Assumption 5.2 that $\min_{\phi \in \Phi} L(\phi) \leq L(\phi^*) \leq \gamma$, along with the above two lemmas completes the proof.

## 6 REPRESENTATION LEARNING FOR OBSERVATION-ALONE SETTING

Now we consider the setting where we cannot observe experts' actions but only their states. As in Sun et al. (2019), we also solve a problem at each level; consider a level $h \in [H]$.

**Choice of $\ell_h^\mu$:** Let $\boldsymbol{\pi}_\mu^* = \{\pi_{1,\mu}^*, \ldots, \pi_{H,\mu}^*\}$ be the sequence of expert policies (possibly stochastic) at different levels for the task $\mu$. Let $\nu_{h,\mu}^*$ be the distribution induced on the states at level $h$ by the expert policy $\boldsymbol{\pi}_\mu^*$. The goal in imitation learning with observations alone (Sun et al., 2019) is

---

[4]These statements are qualitative since we are comparing upper bounds.

to learn a policy $\boldsymbol{\pi} = (\pi_1, \ldots, \pi_H)$ that matches the distributions $\nu_h^{\pi}$ with $\nu_h^*$ for every $h$, w.r.t. a discriminator class $\mathcal{G}$[5] that contains the true value functions $V_1^*, \ldots, V_H^*$ and is *approximately* closed under the Bellman operator of $\boldsymbol{\pi}^*$. Instead, in this work we learn $\boldsymbol{\pi}$ that matches the distributions $\pi_h \cdot \nu_h^*$ and $\nu_{h+1}^*$ for every $h$ w.r.t. to a class $\mathcal{G} \subseteq \{g : \mathcal{S} \to \mathbb{R}, |g|_\infty \le 1\}$ that contains the value functions and has a stronger Bellman operator closure property. For every task $\mu$, $\ell_h^\mu$ is defined as

$$\ell_h^\mu(\pi) = \max_{g \in \mathcal{G}} [\underset{\substack{s \sim \nu_{h,\mu}^* \\ a \sim \pi(s) \\ \tilde{s} \sim P_{s,a}}}{\mathbb{E}} g(\tilde{s}) - \underset{\bar{s} \sim \nu_{h+1,\mu}^*}{\mathbb{E}} g(\bar{s})] \tag{6}$$

$$= \max_{g \in \mathcal{G}} [\underset{\substack{s \sim \nu_{h,\mu}^* \\ a \sim \mathcal{U}(\mathcal{A}) \\ \tilde{s} \sim P_{s,a}}}{\mathbb{E}} K\pi(a|s)g(\tilde{s}) - \underset{\bar{s} \sim \nu_{h+1,\mu}^*}{\mathbb{E}} g(\bar{s})]$$

where we rewrite $\ell_h^\mu$ by importance sampling in the second equation; this will be useful to get an empirical estimate. While our definition of $\ell_h^\mu$ differs slightly from the one used in Sun et al. (2019), using similar techniques, we will show that small values for $\ell_h^\mu(\pi_h)$ for every $h \in [H]$ will ensure that the policy $\boldsymbol{\pi} = (\pi_1, \ldots, \pi_H)$ will have expected cost $J_\mu(\boldsymbol{\pi})$ close to $J_\mu(\boldsymbol{\pi}_\mu^*)$. We abuse notation, and for a task $\mu$ we denote $\mu = (\mu_1, \ldots, \mu_H)$ where $\mu_h$ is the distribution of $(s, a, \tilde{s}, \bar{s})$ used in $\ell_h^\mu$; thus $(s, a, \tilde{s}, \bar{s}) \sim \mu_h$ is equivalent to $s \sim \nu_{h,\mu}^*, a \sim \mathcal{U}(\mathcal{A}), \tilde{s} \sim P_{s,a}, \bar{s} \sim \nu_{h+1,\mu}^*$.

**Learning $\phi_h$ from samples:** We assume, 1) access to $2n$ expert trajectories for $T$ independent *train* tasks, 2) ability to reset the environment at any state $s$ and sample from the transition $P(\cdot|s, a)$ for any $a \in \mathcal{A}$. The second condition is satisfied in many problems equipped with simulators. Using the sampled trajectories for the $T$ tasks $\{\mu^{(1)}, \ldots, \mu^{(T)}\}$ and doing some interaction with environment, we get the following dataset $\mathbf{X} = \{\mathbf{X}_1, \ldots, \mathbf{X}_H\}$ where $\mathbf{X}_h$ is the dataset for level $h$. Specifically, $\mathbf{X}_h = \{\mathbf{x}_h^{(1)}, \ldots, \mathbf{x}_H^{(T)}\}$ where $\mathbf{x}_h^{(i)} = \{(s_j^i, a_j^i, \tilde{s}_j^i, \bar{s}_j^i)\}_{j=1}^n \sim (\mu^{(i)})^n$. Additionally we denote $\mathbf{S}_h = \{s_j^i\}_{i=1,j=1}^{T,n}$ to be all the $s$-states in $\mathbf{X}_h$, $\tilde{\mathbf{S}}_h$ and $\bar{\mathbf{S}}_h$ are similarly defined. Details about how this dataset is constructed from expert trajectories and interactions with environment is provided in Section C.2. We learn the representation $\hat{\phi}_h = \arg\min_{\phi \in \Phi} \hat{L}_h(\phi)$, where

$$\hat{L}_h(\phi) = \frac{1}{T} \sum_{i=1}^T \min_{\pi \in \Pi^\phi} \max_{g \in \mathcal{G}} \frac{1}{n} \sum_{j=1}^n [K\pi(a_j^i|s_j^i)g(\tilde{s}_j^i) - g(\bar{s}_j^i)] = \frac{1}{T} \sum_{i=1}^T \min_{\pi \in \Pi^\phi} \hat{\ell}_h^{\mathbf{x}^{(i)}}(\pi)$$

where for dataset $\mathbf{x} = \{(s_j, a_j, \tilde{s}_j, \bar{s}_j)\}_{j=1}^n$, $\hat{\ell}_h^{\mathbf{x}}(\pi) := \max_{g \in \mathcal{G}} \frac{1}{n} \sum_{j=1}^n [K\pi(a_j|s_j)g(\tilde{s}_j) - g(\bar{s}_j)]$. Note that because of the $\max_{g \in \mathcal{G}}$, $\hat{\ell}_h^{\mathbf{x}}$ is no longer an unbiased estimator of $\ell_h^\mu$ when $\mathbf{x} \sim \mu_h^n$. However we can still show generalization bounds.

**Evaluating representations $\hat{\phi}_1, \ldots, \hat{\phi}_H$:** Learned representations are tested on a new task $\mu \sim \eta$ as follows: get samples $\mathbf{x} = (\mathbf{x}_1, \ldots, \mathbf{x}_H)$[6] for all levels using trajectories from $\boldsymbol{\pi}_\mu^*$, where $\mathbf{x}_h \sim \mu_h^n$. For each level $h$, learn $\pi^{\hat{\phi}_h, \mathbf{x}_h} = \arg\min_{\pi \in \Pi^{\hat{\phi}}} \hat{\ell}_h^{\mathbf{x}_h}(\pi)$ and consider the policy $\boldsymbol{\pi}^{\hat{\phi}, \mathbf{x}} = (\pi^{\hat{\phi}_1, \mathbf{x}_1}, \ldots, \pi^{\hat{\phi}_H, \mathbf{x}_H})$. Before presenting the guarantee for $\boldsymbol{\pi}^{\hat{\phi}, \mathbf{x}}$, we introduce a notion of *Bellman error* that will show up in our results. For a policy $\boldsymbol{\pi} = (\pi_1, \ldots, \pi_H)$ and an expert policy $\boldsymbol{\pi}^* = (\pi_1^*, \ldots, \pi_H^*)$, we define the inherent Bellman error

$$\epsilon_{be}^{\boldsymbol{\pi}} := \max_{h \in [H]} \max_{g \in \mathcal{G}} \min_{g' \in \mathcal{G}} \underset{s \sim (\nu_h^* + \nu_h^{\boldsymbol{\pi}})/2}{\mathbb{E}} [|g'(s) - (\Gamma_h^{\boldsymbol{\pi}} g)(s)|] \tag{7}$$

We make the following two assumptions for the subsequent theorem. These are standard assumptions in theoretical reinforcement learning literature.

**Assumption 6.1** (Value function realizability). $V_{h,\mu}^* \in \mathcal{G}$ *for every* $h \in [H]$, $\mu \in support(\eta)$.

**Assumption 6.2** (Policy realizability). *There are representations* $\phi_1^*, \ldots, \phi_H^* \in \Phi$ *such that* $\pi_{h,\mu}^* \in \Pi^{\phi_h^*}$ *for every* $h \in [H]$, $\mu \in support(\eta)$.

Now we present our main theorem for the observation-alone setting.

---

[5]If $\mathcal{G}$ contains all bounded functions, then it reduces to minimizing TV between $\nu_h^\pi$ and $\nu_h^*$.
[6]Note that we do not need the datasets $\mathbf{x}_h$ at different levels to be independent of each other

**Theorem 6.1.** *Let $\hat{\phi}_h \in \arg\min_{\phi \in \Phi} \hat{L}_h(\phi)$. Under Assumptions 6.1,6.2, with probability $1 - \delta$ over sampling of $\mathbf{X} = (\mathbf{X}_1, \ldots, \mathbf{X}_H)$, we have*

$$\mathbb{E}_{\mu \sim \eta} \mathbb{E}_{\mathbf{x}} J(\boldsymbol{\pi}^{\hat{\phi}, \mathbf{x}}) - \mathbb{E}_{\mu \sim \eta} J(\boldsymbol{\pi}_\mu^*) \leq \sum_{h=1}^{H} (2H - 2h + 1)\epsilon_{gen,h} + O(H^2)\epsilon_{be}^{\hat{\phi}}$$

*where $\epsilon_{be}^{\hat{\phi}} = \mathbb{E}_{\mu \sim \eta} \mathbb{E}_{\mathbf{x}}[\epsilon_{be}^{\boldsymbol{\pi}^{\hat{\phi}, \mathbf{x}}}]$ is the average inherent Bellman error and*

$$\epsilon_{gen,h} = O\left( \frac{KG(\Phi(\mathbf{S}_h))}{T\sqrt{n}} + \mathbb{E}_{\mu \sim \eta} \mathbb{E}_{\mathbf{x} \sim \mu^n} \left[ \frac{KG(\mathcal{G}(\tilde{\mathbf{s}}_h))}{n} + \frac{G(\mathcal{G}(\bar{\mathbf{s}}_h))}{n} \right] + \frac{RK\sqrt{K}}{\sqrt{n}} + \sqrt{\frac{\ln(H/\delta)}{T}} \right)$$

We again give a PAC-style guarantee for the special case where the class of representation functions $\Phi$ and value function class $\mathcal{G}$ are finite. It follows from the above theorem and Massart's lemma.

**Corollary 6.1.** *In the setting of Theorem 6.1, suppose $\Phi, \mathcal{G}$ are finite. If number of tasks satisfies $T \geq c_1 \max\left\{ \frac{H^4 R^2 K^2 \log(|\Phi|)}{\epsilon^2}, \frac{H^4 \ln(H/\delta)}{\epsilon^2} \right\}$, and number of samples (trajectories) per task satisfies $n \geq c_2 \max\left\{ \frac{H^4 K^2 \log(|\mathcal{G}|)}{\epsilon^2}, \frac{H^4 R^2 K^3}{\epsilon^2} \right\}$ for small constants $c_1, c_2$, then with probability $1 - \delta$,*

$$\mathbb{E}_{\mu \sim \eta} \mathbb{E}_{\mathbf{x}} J(\boldsymbol{\pi}^{\hat{\phi}, \mathbf{x}}) - \mathbb{E}_{\mu \sim \eta} J(\boldsymbol{\pi}_\mu^*) \leq O(H^2)\epsilon_{be}^{\hat{\phi}} + \epsilon.$$

**Discussion:** As in the previous section, the number of samples required for a new task after learning a representation is independent of the class $\Phi$ but depends only on the value function class $\mathcal{G}$ and number of actions. Thus representation learning is very useful when the class $\Phi$ is much more complicated than $\mathcal{G}$, i.e. $R^2 \log(|\Phi|) \gg \max\{\log(|\mathcal{G}|), R^2 K\}$. In the above bounds, $\epsilon_{be}^{\hat{\phi}}$ is a Bellman error term. This type of error terms occur commonly in the analysis of policy iteration type algorithms (Munos, 2005; Munos & Szepesvári, 2008). We remark that unlike in Sun et al. (2019), our Bellman error is based on the Bellman operator of the learned policy rather than the optimal policy. Le et al. (2019) used a similar notion that they call *inherent Bellman evaluation error*.

The proof of Theorem 6.1 follows a similar outline to that of behavioral cloning. However we cannot use the results from Maurer et al. (2016) directly since we are solving a min-max game for each task. We provide the proof in Appendix B.

## 7 EXPERIMENTS

In this section we present experimental results on the DirectedSwimmer environment (modified from the Swimmer environment from OpenAI gym (Brockman et al., 2016)) with Todorov et al. (2012) simulator and a NoisyCombinationLock environment designed by ourself. These experiments have two aims: 1) verify the benefit of representation learning predicted by our theory, 2) test the power of representations learned via our framework in a broader context: we learn a policy for a new task by using the representation and doing policy optimization instead of imitation learning. In our experiments we learn representations using Equation 5. Experiment details are deferred to Section D.

**Our method:** Given access to a dataset $\mathbf{X} = \{(s_j^t, a_j^t)\}_{j=1}^{n}$ of $n$ state-action pairs each for $T$ tasks, we learn a $\hat{\phi}$ according to Equation 8. For any new task we learn a linear policy $\pi$ from the class $\Pi^{\hat{\phi}}$.

$$\hat{\phi} = \arg\min_{\phi \in \Phi} \min_{f_1, \ldots, f_T \in \mathcal{F}} \frac{1}{T} \sum_{t=1}^{T} \frac{1}{n} \sum_{j=1}^{n} -\log(\pi^{\phi, f_t}(s_j^t)_{a_j^t}) \tag{8}$$

**Baseline:** For a task we learn a policy $\pi$ from the class $\Pi$ without learning a representation first.

**Verification of theory:** In Figure 1 we verify our theoretical findings. On the left, we test on the DirectedSwimmer environment and report the logistic loss on the validation, which measures how close the trained policy is to the target expert policy. We find that learning representations, even with a few experts, can significantly reduce the sample complexity. On the right, we report the average

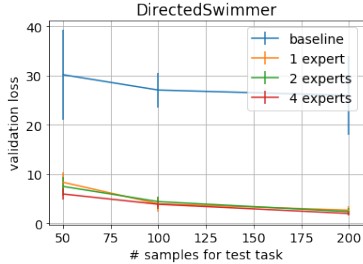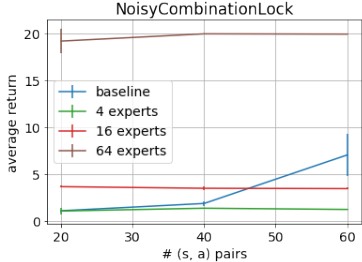

Figure 1: Experiments for verifying theory. Left: validation loss on DirectedSwimmer. Right: average return on NoisyCombinationLock

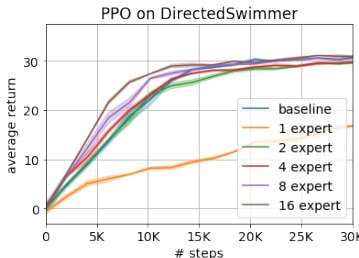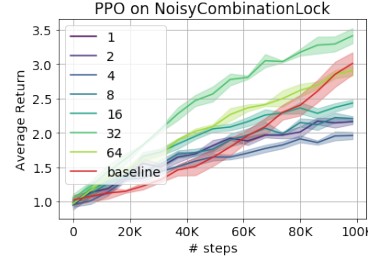

Figure 2: Experiments on policy Optimization with representation trained by imitation learning Left: average return on the DirectedSwimmer. Right: average return on the NoisyCombinationLock.

reward of the trained policies on the environment. Here we see a different phenomenon: when the number of experts is small (4 or 16), the baseline method can beat policies trained using representation learning, though the baseline method requires more samples to do so. When the number of experts is large (64), we see the policy trained using representation learning can significantly outperform the baseline method. This behavior is expected as when the number of experts is small, we may learn a sub-optimal representation and because we fix this representation for training the policy, more samples for the test task cannot make this policy better, whereas more samples always make the baseline method better. Nevertheless, when the number of experts is large, we can significantly reduce the sample complexity. With 60 samples, the base line method is still far behind the policy trained using representation learning with 64 experts.

**Policy optimization with representations trained by imitation learning:** We next test the utility of representations learned via our framework for RL. After training a representation, we use a simplified proximal policy optimization method that learns a linear policy over the learned representation. Results are reported in Figure 2. For DirectedSwimmer and NoisyCombinationLock, we observe a common pattern. When the number of experts to learn the representation is small, the baseline method enjoys better performance than the policies trained using representation learning. As the number of experts to learn the representation increases, we see the policy trained using representation learning can initially outperform baseline, sometime significantly. However, unsurprisingly, the baseline method performs very well with a large number of samples, since it is allowed to learn a representation from scratch. This experiment suggests that representations trained via imitation learning can be useful *beyond imitation learning*, especially when the target task has few samples.

# 8 CONCLUSION

The current paper proposes a bi-level optimization framework to formulate and analyze representation learning for imitation learning using multiple demonstrators. Theoretical guarantees are provided to justify the statistical benefit of representation learning. Some preliminary experiments verify the effectiveness of the proposed framework. In particular, in experiments, we find the representation learned via imitation learning is also useful for policy optimization in the reinforcement learning setting. We believe it is an interesting theoretical question to explain this phenomenon. Additionally, extending this bi-level optimization framework to incorporate methods beyond imitation learning is an interesting future direction.

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

## A  PROOFS FOR BEHAVIORAL CLONING

We prove Theorem 5.1 in this section by proving Lemma 5.2,5.3. In this section, we abuse notation and define $\ell^\mu(\phi, f) := \ell^\mu(\pi^{\phi,f})$, where $\ell^\mu$ is defined in Equation 4. Let $\hat{f}_{\mathbf{x}}^\phi = \arg\min_{f \in \mathcal{F}} \ell^{\mathbf{x}}(\phi, f)$ be the optimal task specific parameter for task $\mu$ by fixing representation $\phi$. Thus by our definitions in Section 5, we get $\pi^{\phi,\mathbf{x}} = \pi^{\phi,\hat{f}_{\mathbf{x}}^\phi}$. We assume w.l.o.g. that $\mathcal{A} = [K]$. Remember that $\ell : \triangle(\mathcal{A}) \times \mathcal{A} \to \mathbb{R}$ is defined as $\ell(\boldsymbol{v}, a) = -\log(\boldsymbol{v}_a)$ for some $\boldsymbol{v} \in \mathbb{R}^K$ and $\boldsymbol{v}_a$ is the coordinate corresponding to action $a \in \mathcal{A} = [K]$. We define a new function class and loss function that will be useful for our proofs

$$\mathcal{F}' = \{x \to Wx \mid W \in \mathbb{R}^{K \times d}, \|W\|_F \leq 1\} \tag{9}$$

$$\ell'(\boldsymbol{v}, a) = -\log(\texttt{softmax}(\boldsymbol{v})_a), \boldsymbol{v} \in \mathbb{R}^K, a \in \mathcal{A} \tag{10}$$

We basically offloaded the burden of computing $\texttt{softmax}$ from the class $\mathcal{F}$ to the loss $\ell$. We can convert any function $f' \in \mathcal{F}'$ to one in $\mathcal{F}$ by transforming it to $\texttt{softmax}(f')$.

We now proceed to proving the lemmas

*Proof of Lemma 5.2.*  We can then rewrite the various loss functions from Section 5 as follows

$$\hat{L}(\phi) = \frac{1}{T} \sum_{i=1}^{T} \min_{f \in \mathcal{F}'} \frac{1}{n} \sum_{j=1}^{n} \ell'(f(\phi(s)), a)$$

$$L(\phi) = \mathop{\mathbb{E}}_{\mu \sim \eta} \min_{f \in \mathcal{F}'} \mathop{\mathbb{E}}_{(s,a) \sim \mu} \ell'(f(\phi(s)), a)$$

$$\bar{L}(\phi) = \mathop{\mathbb{E}}_{\mu \sim \eta} \mathop{\mathbb{E}}_{\mathbf{x} \sim \mu^n} \mathop{\mathbb{E}}_{(s,a) \sim \mu} \ell'(\hat{f'}_{\mathbf{x}}^\phi(\phi(s)), a)$$

where $\hat{f'}_\mu^\phi \in \arg\min_{f' \in \mathcal{F}'} \ell^{\mathbf{x}}(\phi, \texttt{softmax}(f'))$. It is easy to show that both $\ell'(\cdot, a)$ $\ell'(f'(\cdot), \cdot)$ are 2-lipschitz in their arguments for every $a \in \mathcal{A}$ and $f' \in \mathcal{F}'$. Using a slightly modified version of Theorem 2(i) from Maurer et al. (2016), we get that for $\hat{\phi} \in \arg\min_{\phi \in \Phi} \hat{L}(\phi)$, with probability at least $1 - \delta$ over the choice of $\mathbf{X}$

$$\bar{L}(\hat{\phi}) - \min_{\phi \in \Phi} L(\phi) \leq \frac{2\sqrt{2\pi}G(\Phi(\mathbf{S}))}{T\sqrt{n}} + \sqrt{2\pi}Q' \sup_{\phi \in \Phi} \sqrt{\frac{\mathop{\mathbb{E}}_{\mu \sim \eta, (s,a) \sim \mu} \|\phi(s)\|^2}{n}} + \sqrt{\frac{8\log(4/\delta)}{T}}$$

$$\bar{L}(\hat{\phi}) - \min_{\phi \in \Phi} L(\phi) \leq c\frac{G(\Phi(\mathbf{S}))}{T\sqrt{n}} + c'\frac{Q'R}{\sqrt{n}} + c''\sqrt{\frac{\log(4/\delta)}{T}} \tag{11}$$

where $Q' = \sup_{y \in \mathbb{R}^{dn}\setminus\{0\}} \frac{1}{\|y\|}\mathbb{E} \sup_{f \in \mathcal{F}'} \sum_{i=1,j=1}^{n,K} \gamma_{ij}f'(y_i)_j$. First we discuss why we need a modified version of their theorem. Our setting differs from the setting for Theorem 2 from Maurer et al. (2016) in the following ways

- $\mathcal{F}'$ is a class of vector valued function in our case, whereas in Maurer et al. (2016) it is assumed to contain scalar valued. The only place in the proof of the theorem where this shows up is in the definition of $Q'$, which we have updated accordingly.
- Maurer et al. (2016) assumes that $\ell'(\cdot, a)$ is 1-lipschitz for every $a \in \mathcal{A}$ and that $f'(\cdot)$ is $L$ lipschitz for every $f' \in \mathcal{F}'$. However the only properties that are used in the proof of Theorem 16 are that $\ell'(\cdot, a)$ is 1-lipschitz and that $\ell'(f'(\cdot), a)$ is $L$-lipschitz for every $a \in \mathcal{A}$, which is exactly the property that we have. Hence their proof follows through for our setting as well.

**Lemma A.1.** $Q' := \sup_{y \in \mathbb{R}^{dn}\setminus\{0\}} \frac{1}{\|y\|}\mathbb{E} \sup_{f \in \mathcal{F}'} \sum_{i=1,j=1}^{n,K} \gamma_{ij}f'(y_i)_j \leq \sqrt{K}$

*Proof.*

$$Q' := \sup_{y \in \mathbb{R}^{dn} \setminus \{0\}} \frac{1}{\|y\|} \mathbb{E} \sup_{f \in \mathcal{F}'} \sum_{i=1,j=1}^{n,K} \gamma_{ij} f'(y_i)_j$$

$$= \sup_{y \in \mathbb{R}^{dn} \setminus \{0\}} \frac{1}{\|y\|} \mathbb{E} \sup_{\|W\|_F \leq 1} \sum_{i=1,j=1}^{n,K} \gamma_{ij} \langle W_j, y_i \rangle$$

$$= \sup_{y \in \mathbb{R}^{dn} \setminus \{0\}} \frac{1}{\|y\|} \mathbb{E} \sup_{\|W\|_F \leq 1} \sum_{j=1}^{K} \langle W_j, \sum_{i=1}^{n} \gamma_{ij} y_i \rangle$$

$$= \sup_{y \in \mathbb{R}^{dn} \setminus \{0\}} \frac{1}{\|y\|} \mathbb{E} \sqrt{\sum_{j=1}^{K} \left\| \sum_{i=1}^{n} \gamma_{ij} y_i \right\|^2}$$

$$\leq \sup_{y \in \mathbb{R}^{dn} \setminus \{0\}} \frac{1}{\|y\|} \sqrt{\sum_{j=1}^{K} \mathbb{E} \left\| \sum_{i=1}^{n} \gamma_{ij} y_i \right\|^2}$$

$$= \sup_{y \in \mathbb{R}^{dn} \setminus \{0\}} \frac{1}{\|y\|} \sqrt{\sum_{j=1}^{K} \sum_{i=1}^{n} \|y_i\|^2} = \frac{1}{\|y\|} \sqrt{K \|y\|^2} = \sqrt{K}$$

where we use Jensen's inequality and linearity of expectation for the first inequality and properties of standard normal gaussian variables for the equality after that. □

Plugging in Lemma A.1 into Equation 11 completes the proof. □

We now proceed to prove the next lemma.

*Proof of Lemma 5.3.* Suppose $\bar{L}(\phi) = \mathbb{E}_{\mu \sim \eta} \mathbb{E}_{\mathbf{x} \sim \mu^n} \ell^\mu(\pi^{\phi, \mathbf{x}}) \leq \epsilon$. Consider a task $\mu \sim \eta$ and samples $\mathbf{x} \sim \mu^n$ and let $\epsilon_\mu(\mathbf{x}) = \ell^\mu(\pi^{\phi, \mathbf{x}})$ so that $\bar{L}(\phi) = \mathbb{E}_{\mu \sim \eta} \mathbb{E}_{\mathbf{x} \sim \mu^n} \epsilon_\mu(\mathbf{x})$. Since $\pi_\mu^*$ is deterministic, we get

$$\mathbb{E}_{s \sim \nu_\mu^*} \mathbb{E}_{a \sim \pi^{\phi, \mathbf{x}}} \mathbb{I}\{a \neq \pi_\mu^*(s)\} = \mathbb{E}_{s \sim \nu_\mu^*} [1 - \pi^{\phi, \mathbf{x}}(s)_{\pi_\mu^*(s)}]$$

$$\leq \mathbb{E}_{s \sim \nu_\mu^*} [-\log(1 - (1 - \pi^{\phi, \mathbf{x}}(s)_{\pi_\mu^*(s)}))]$$

$$= \mathbb{E}_{s \sim \nu_\mu^*} [-\log(\pi^{\phi, \mathbf{x}}(s)_{\pi_\mu^*(s)})] = \epsilon_\mu(\mathbf{x})$$

where we use the fact that $x \leq -\log(1 - x)$ for $x < 1$. for the first inequality. Thus by using Theorem 2.1 from Ross et al. (2011), we get that $J_\mu(\pi^{\phi, \mathbf{x}}) - J_\mu(\pi^*) \leq H^2 \epsilon_\mu(\mathbf{x})$. Taking expectation w.r.t. $\mu \sim \eta$ and $\mathbf{x} \sim \mu^n$ completes the proof. □

*Proof of Theorem 5.1.* By using Assumption 5.2, we first get that

$$L(\phi^*) = \mathbb{E}_{\mu \sim \eta} \min_{\pi \in \Pi^{\phi^*}} \mathbb{E}_{s \sim \nu_\mu^*} -\log(\pi(s)_{\pi_\mu^*(s)})$$

$$\leq \mathbb{E}_{\mu \sim \eta} \mathbb{E}_{s \sim \nu_\mu^*} -\log(\pi_\mu(s)_{\pi_\mu^*(s)})$$

$$\leq \mathbb{E}_{\mu \sim \eta} \mathbb{E}_{s \sim \nu_\mu^*} -\log(1 - \gamma) \leq 2\gamma$$

where in the last step we used $-\log(1 - x) \leq 2x$ for $x < 1/2$. Hence from Lemma 5.2 we get $\bar{L}(\hat{\phi}) \leq 2\gamma + \epsilon_{gen,h}$, which combining with Lemma 5.3 gives the desired result. □

# B PROOFS FOR OBSERVATION-ALONE

Before proving Theorem 6.1, we introduce the following loss functions, as we did in the proof sketch for the behavioral cloning setting. We again abuse notation and define $\ell^\mu(\phi, f) := \ell^\mu(\pi^{\phi,f})$, where $\ell^\mu$ is defined in Equation 6. Let $\hat{f}_{\mathbf{x}}^\phi = \arg\min_{f \in \mathcal{F}} \ell^{\mathbf{x}}(\phi, f)$ be the optimal task specific parameter for task $\mu$ by fixing representation $\phi$. As before, we define the following

$$\bar{L}_h(\phi_h) = \mathop{\mathbb{E}}_{\mu \sim \eta} \mathop{\mathbb{E}}_{\mathbf{x} \sim \mu_h^n} \ell_h^\mu(\phi, \hat{f}_{\mathbf{x}}^{\phi_h})$$

We first show a guarantee on the performance of representations $(\hat{\phi}_1, \ldots, \hat{\phi}_H)$ as measured by the functions $\bar{L}_1, \ldots, \bar{L}_H$.

**Theorem B.1.** *With probability at least $1 - \delta$ in the draw of $\mathbf{X} = (\mathbf{X}^{(1)}, \ldots, \mathbf{X}^{(H)})$, $\forall h \in [H]$*

$$\bar{L}_h(\hat{\phi}_h) \leq \min_{\phi \in \Phi} L_h(\phi) + c\epsilon_{gen,h}(\Phi) + c'\epsilon_{gen,h}(\mathcal{F}, \mathcal{G}) + c''\sqrt{\frac{\ln(H/\delta)}{T}}$$

*where $\epsilon_{gen,h}(\Phi) = \frac{KG(\Phi(\mathbf{S}_h))}{T\sqrt{n}}$ and $\epsilon_{gen,h}(\mathcal{F}, \mathcal{G}) = \mathop{\mathbb{E}}_{\mu \sim \eta} \mathop{\mathbb{E}}_{\mathbf{x} \sim \mu^n} \left[ \frac{KG(\mathcal{G}(\bar{\mathbf{s}}_h))}{n} + \frac{G(\mathcal{G}(\bar{\mathbf{s}}_h))}{n} \right] + \frac{RK\sqrt{K}}{\sqrt{n}}$*

We then connect the losses $\bar{L}_h$ to the expected cost on the tasks.

**Theorem B.2.** *Consider representations $(\phi_1, \ldots, \phi_H)$ with $\bar{L}_h(\phi_h) \leq \epsilon_h$. Let $\mathbf{x} = (\mathbf{x}_1, \ldots, \mathbf{x}_H)$ be samples at different levels for a newly sampled task $\mu \sim \eta$ such that $\mathbf{x}_h \sim \mu_h^n$. Let $\boldsymbol{\pi}^{\phi, \mathbf{x}} = (\pi^{\phi_1, \mathbf{x}_1}, \ldots, \pi^{\phi_H, \mathbf{x}_H})$ be policies learned using the samples, then under Assumption 6.1,*

$$\mathop{\mathbb{E}}_{\mu \sim \eta} \mathop{\mathbb{E}}_{\mathbf{x}} J(\boldsymbol{\pi}^{\phi, \mathbf{x}}) - \mathop{\mathbb{E}}_{\mu \sim \eta} J(\boldsymbol{\pi}_\mu^*) \leq \sum_{h=1}^{H} (2H - 2h + 1)\epsilon_h + O(H^2)\epsilon_{be}^\phi$$

*where $\epsilon_{be}^\phi = \mathop{\mathbb{E}}_{\mu \sim \eta} \mathop{\mathbb{E}}_{\mathbf{x}}[\epsilon_{be}^{\boldsymbol{\pi}^{\phi, \mathbf{x}}}]$ is the average inherent Bellman error.*

It is easy to show that under Assumption 6.2, $\min_{\phi \in \Phi} L_h(\phi) = 0$ for every $h \in [H]$. Thus from Theorem B.1, we get that $\bar{L}_h(\hat{\phi}_h) \leq \epsilon_{gen,h}$, where $\epsilon_{gen,h} = \epsilon_{gen,h}(\Phi) + \epsilon_{gen,h}(\mathcal{F}, \mathcal{G}) + c''\sqrt{\frac{\ln(H/\delta)}{T}}$. Invoking Theorem B.2 on the representations $\{\hat{\phi}_h\}$ completes the proof.

## B.1 PROOF OF THEOREM B.1

Before proving the theorem, we discuss important lemmas. In yet another abuse of notation, we define $\ell_h^\mu(\phi, f, g) = \mathbb{E}_{(s,a,\tilde{s},\bar{s}) \sim \mu_h}[K\pi^{\phi,f}(a|s)g(\tilde{s}) - g(\bar{s})]$ and $\ell_h^{\mathbf{x}}(\phi, f, g) = \frac{1}{n} \sum_{j=1}^{n} [K\pi^{\phi,f}(a_j|s_j)g(\tilde{s}_j) - g(\bar{s}_j)]$.

Let $\hat{m}_{\mathbf{x}}(\phi) = \min_{f \in \mathcal{F}} \max_{g \in \mathcal{G}} \hat{\ell}_h^{\mathbf{x}}(\phi, f, g) = \hat{\ell}_h^{\mathbf{x}}(\phi, \hat{f}_{\mathbf{x}}^\phi, \hat{g}_{\mathbf{x}}^\phi)$, $\bar{m}_{\mu, \mathbf{x}}(\phi) = \max_{g \in \mathcal{G}} \ell_h^\mu(\phi, \hat{f}_{\mathbf{x}}^\phi, g)$, $m_\mu(\phi) = \min_{f \in \mathcal{F}} \max_{g \in \mathcal{G}} \ell_h^\mu(\phi, f, g)$. Note that $L_h(\phi) = \mathop{\mathbb{E}}_{\mu \sim \eta} m(\phi)$, $\bar{L}_h(\phi) = \mathop{\mathbb{E}}_{\mu \sim \eta} \mathop{\mathbb{E}}_{\mathbf{x} \sim \mu^n} \bar{m}_{\mu, \mathbf{x}}(\phi)$. Define the distribution $\rho_h$ where $\mathbf{x} \sim \rho_h$ is the same as $\mu \sim \eta$ and then $\mathbf{x} \sim \mu_h^n$.

**Lemma B.3.** *For every $\phi \in \Phi$ and $h \in [H]$,*

$$\mathop{\mathbb{E}}_{\mu \sim \eta} \mathop{\mathbb{E}}_{\mathbf{x} \sim \mu^n} \sup_{f \in \mathcal{F}} \sup_{g \in \mathcal{G}} \left[ \hat{\ell}_h^{\mathbf{x}}(\phi, f, g) - \ell_h^\mu(\phi, f, g) \right] \leq \epsilon_{gen,h}(\mathcal{F}, \mathcal{G})$$

**Lemma B.4.** *With probability $1 - \delta$, for every $\phi \in \Phi$,*

$$\bar{L}_h(\phi) - \mathop{\mathbb{E}}_{\mathbf{x} \sim \rho_h} \hat{m}_{\mathbf{x}}(\phi) \leq \epsilon_{gen,h}(\mathcal{F}, \mathcal{G})$$

**Lemma B.5.** *With probability $1 - \delta$, for every $\phi \in \Phi$,*

$$\mathop{\mathbb{E}}_{\mathbf{x} \sim \rho_h} \hat{m}_{\mathbf{x}}(\phi) - \frac{1}{T} \sum_i \hat{m}_{\mathbf{x}^{(i)}}(\phi) \leq \epsilon_{gen,h}(\Phi) + O\left( \sqrt{\frac{\log(\frac{1}{\delta})}{T}} \right)$$

We prove these lemmas later. First we prove Theorem B.1 using them. If $\phi_h^* = \arg\min\limits_{\phi \in \Phi} L_h(\phi)$, then

$$
\begin{aligned}
\bar{L}_h(\hat{\phi}_h) - L_h(\phi_h^*) = &\left( \bar{L}_h(\hat{\phi}_h) - \mathop{\mathbb{E}}\limits_{\mathbf{x} \sim \rho_h} \hat{m}_{\mathbf{x}}(\phi) \right) \\
&+ \left( \mathop{\mathbb{E}}\limits_{\mathbf{x} \sim \rho_h} \hat{m}_{\mathbf{x}}(\phi) - \frac{1}{T} \sum_i \hat{m}_{\mathbf{x}^{(i)}}(\hat{\phi}_h) \right) \\
&+ \left( \frac{1}{T} \sum_i \hat{m}_{\mathbf{x}^{(i)}}(\hat{\phi}_h) - \frac{1}{T} \sum_i \hat{m}_{\mathbf{x}^{(i)}}(\phi_h^*) \right) \\
&+ \left( \frac{1}{T} \sum_i \hat{m}_{\mathbf{x}^{(i)}}(\phi_h^*) - \mathop{\mathbb{E}}\limits_{\mathbf{x} \sim \rho_h} \hat{m}_{\mathbf{x}}(\phi_h^*) \right) \\
&+ \mathop{\mathbb{E}}\limits_{\mu \sim \eta} [\mathop{\mathbb{E}}\limits_{\mathbf{x} \sim \mu^n} \hat{m}_{\mathbf{x}}(\phi_h^*) - m_\mu(\phi_h^*)] \\
\leq\; &2\epsilon_{gen,h}(\mathcal{F}, \mathcal{G}) + \epsilon_{gen,h}(\Phi) + O\left( \sqrt{\frac{\log(\frac{1}{\delta})}{T}} \right)
\end{aligned}
$$

where for the first part we use Lemma B.4, second part we use Lemma B.5, third part is upper bounded by 0 by optimality of $\hat{\phi}_h$, fourth is upper bounded by $O(\sqrt{\frac{\log(\frac{1}{\delta})}{T}})$ by Hoeffding's inequality and fifth is bounded by the following argument: let $f^\phi, g^\phi = \arg\min\limits_{f \in \mathcal{F}} \arg\max\limits_{g \in \mathcal{G}} \ell^\mu(\phi, f, g)$

$$
\begin{aligned}
\mathop{\mathbb{E}}\limits_{\mathbf{x} \sim \mu^n} \hat{m}_{\mathbf{x}}(\phi_h^*) = &\mathop{\mathbb{E}}\limits_{\mathbf{x} \sim \mu^n} \min\limits_{f \in \mathcal{F}} \max\limits_{g \in \mathcal{G}} \hat{\ell}_h^{\mathbf{x}}(\phi_h^*, f, g) \\
\leq\; &\mathop{\mathbb{E}}\limits_{\mathbf{x} \sim \mu^n} \max\limits_{g \in \mathcal{G}} \hat{\ell}_h^{\mathbf{x}}(\phi_h^*, f^{\phi_h^*}, g) = \mathop{\mathbb{E}}\limits_{\mathbf{x} \sim \mu^n} \hat{\ell}_h^{\mathbf{x}}(\phi_h^*, f^{\phi_h^*}, \tilde{g}) \\
\leq\; &\ell_h^\mu(\phi_h^*, f^{\phi_h^*}, \tilde{g}) + \epsilon_{gen,h}(\mathcal{F}, \mathcal{G}) \\
\leq\; &\ell_h^\mu(\phi_h^*, f^{\phi_h^*}, g^{\phi_h^*}) + \epsilon_{gen,h}(\mathcal{F}, \mathcal{G}) = m_\mu(\phi_h^*) + \epsilon_{gen,h}(\mathcal{F}, \mathcal{G})
\end{aligned}
$$

where the second inequality uses Lemma B.3.

## B.2 PROOF OF THEOREM B.2

Consider a task $\mu$. For simplicity of notation, we use $\pi_h$ instead $\pi^{\phi_h, \mathbf{x}_h}$, $\pi$ instead of $\pi^{\phi, \mathbf{x}}$. Let $\nu_h^\pi$ and $\nu_h^*$ be the state distributions at level $h$ induced by $\pi^{\phi, \mathbf{x}}$ and $\pi_\mu^*$ respectively. Let

$$
\epsilon_h(\mathbf{x}_h) = \max\limits_{g \in \mathcal{G}} \mathop{\mathbb{E}}\limits_{s \sim \nu_h^*} [\mathop{\mathbb{E}}\limits_{\substack{a \sim \pi_h \\ s' \sim P_{s,a}}} g(s') - \mathop{\mathbb{E}}\limits_{\substack{a \sim \pi_h^* \\ s' \sim P_{s,a}}} g(s')]
$$

be the loss of policy $\pi_h$ at level $h$. By definition, $\epsilon_h = \mathop{\mathbb{E}}\limits_{\mu \sim \eta} \mathop{\mathbb{E}}\limits_{\mathbf{x} \sim \mu_h^n} \epsilon_h(\mathbf{x})$. Using Lemma C.1 from Sun et al. (2019), we have

$$
J(\pi^{\phi, \mathbf{x}}) - J(\pi_\mu^*) = \sum_{h=1}^H \bar{\Delta}_h = \sum_{h=1}^H \mathop{\mathbb{E}}\limits_{s \sim \nu_h^\pi} \left[ \mathop{\mathbb{E}}\limits_{a \sim \pi_h(\cdot|s), s' \sim P_{s,a}} V_{h+1}^*(s') - \mathop{\mathbb{E}}\limits_{a \sim \pi_h^*(\cdot|s), s' \sim P_{s,a}} V_{h+1}^*(s') \right]
$$

Observe that

$$
\begin{aligned}
\bar{\Delta}_h = &\mathop{\mathbb{E}}\limits_{s \sim \nu_h^\pi} [\mathop{\mathbb{E}}\limits_{a \sim \pi_h(\cdot|s), s' \sim P_{s,a}} V_{h+1}^*(s') - \mathop{\mathbb{E}}\limits_{a \sim \pi_h^*(\cdot|s), s' \sim P_{s,a}} V_{h+1}^*(s')] \\
\leq\; &\max\limits_{g \in \mathcal{G}} \mathop{\mathbb{E}}\limits_{s \sim \nu_h^*} [\mathop{\mathbb{E}}\limits_{a \sim \pi_h(\cdot|s), s' \sim P_{s,a}} g(s') - \mathop{\mathbb{E}}\limits_{a \sim \pi_h^*(\cdot|s), s' \sim P_{s,a}} g(s')] + \\
&\max\limits_{g \in \mathcal{G}} [\mathop{\mathbb{E}}\limits_{s \sim \nu_h^\pi} \Gamma_h^\pi g(s) - \mathop{\mathbb{E}}\limits_{s \sim \nu_h^*} \Gamma_h^\pi g(s)] + [\mathop{\mathbb{E}}\limits_{s \sim \nu_h^\pi} \Gamma_h^* V_{h+1}^*(s) - \mathop{\mathbb{E}}\limits_{s \sim \nu_h^\pi} \Gamma_h^* V_{h+1}^*(s)] \\
\leq\; &\epsilon_h(\mathbf{x}_h) + \max\limits_{g \in \mathcal{G}} [\mathop{\mathbb{E}}\limits_{s \sim \nu_h^\pi} \Gamma_h^\pi g(s) - \mathop{\mathbb{E}}\limits_{s \sim \nu_h^*} \Gamma_h^\pi g(s)] + \max\limits_{g \in \mathcal{G}} [\mathop{\mathbb{E}}\limits_{s \sim \nu_h^\pi} g(s) - \mathop{\mathbb{E}}\limits_{s \sim \nu_h^*} g(s)]
\end{aligned}
$$

**Lemma B.6.** *Defining* $\Delta_h = \max\limits_{g \in \mathcal{G}} \left| \mathop{\mathbb{E}}\limits_{s \sim \nu_h^\pi} g(s) - \mathop{\mathbb{E}}\limits_{s \sim \nu_h^*} g(s) \right|$, *we have*

$$\max_{g \in \mathcal{G}} [ \mathop{\mathbb{E}}_{s \sim \nu_h^\pi} \Gamma_h^{\boldsymbol{\pi}} g(s) - \mathop{\mathbb{E}}_{s \sim \nu_h^*} \Gamma_h^{\boldsymbol{\pi}} g(s) ] \leq \Delta_h + 2\epsilon_{be}^{\boldsymbol{\pi}}$$

Using the above lemma, we get $\bar{\Delta}_h \leq \epsilon_h(\mathbf{x}_h) + 2\Delta_h + 2\epsilon_{be}^{\boldsymbol{\pi}}$. We now bound $\Delta_h$

$$\Delta_h = \max_{g \in \mathcal{G}} \left| \mathop{\mathbb{E}}_{s \sim \nu_{h-1}^\pi} \mathop{\mathbb{E}}_{\substack{a \sim \pi_{h-1} \\ s' \sim P_{s,a}}} g(s') - \mathop{\mathbb{E}}_{s \sim \nu_h^*} g(s) \right|$$

$$= \max_{g \in \mathcal{G}} \left| \mathop{\mathbb{E}}_{s \sim \nu_{h-1}^\pi} \mathop{\mathbb{E}}_{\substack{a \sim \pi_{h-1} \\ s' \sim P_{s,a}}} g(s') - \mathop{\mathbb{E}}_{s \sim \nu_{h-1}^*} \mathop{\mathbb{E}}_{\substack{a \sim \pi_{h-1} \\ s' \sim P_{s,a}}} g(s) \right| + \max_{g \in \mathcal{G}} \left| \mathop{\mathbb{E}}_{s \sim \nu_{h-1}^*} \mathop{\mathbb{E}}_{\substack{a \sim \pi_{h-1} \\ s' \sim P_{s,a}}} g(s') - \mathop{\mathbb{E}}_{s \sim \nu_h^*} g(s) \right|$$

$$= \max_{g \in \mathcal{G}} \left| \mathop{\mathbb{E}}_{s \sim \nu_{h-1}^\pi} \Gamma_{h-1}^{\boldsymbol{\pi}} g(s') - \mathop{\mathbb{E}}_{s \sim \nu_{h-1}^*} \Gamma_{h-1}^{\boldsymbol{\pi}} g(s) \right| + \epsilon_{h-1}(\mathbf{x}_{h-1})$$

$$\leq \Delta_{h-1} + 2\epsilon_{be}^{\boldsymbol{\pi}} + \epsilon_{h-1}(\mathbf{x}_{h-1})$$

Thus $\Delta_h \leq 2(h-1)\epsilon_{be}^{\boldsymbol{\pi}} + \epsilon_{1:h-1}(\mathbf{x}_{1:h-1})$ and so $\bar{\Delta}_h \leq \epsilon_{1:h}(\mathbf{x}_{1:h}) + \epsilon_{1:h-1}(\mathbf{x}_{1:h-1}) + (4h-2)\epsilon_{be}^{\boldsymbol{\pi}}$. This implies that

$$J(\boldsymbol{\pi}^{\phi,\mathbf{x}}) - J(\boldsymbol{\pi}^*) = \sum_{h=1}^H \bar{\Delta}_h \leq \sum_{h=1}^H (2H - 2h + 1)\epsilon_h(\mathbf{x}_h) + O(H^2)\epsilon_{be}^{\boldsymbol{\pi}^{\phi,\mathbf{x}}}$$

Taking expectation wrt $\mu \sim \eta$ and $\mathbf{x} \sim \mu^n$ completes the proof.

### B.3 PROOFS OF LEMMAS

*Proof of Lemma B.3.* Again we define $\mathcal{F}'$ as in Equation 9. Let $\ell(\boldsymbol{v}, \alpha, \beta, a) = K\,\texttt{softmax}(\boldsymbol{v})_a \alpha - \beta$, and let $\ell_h'^\mu(\phi, f', g) = \ell_h'^\mu(\phi, \texttt{softmax}(f'), g) = \mathop{\mathbb{E}}\limits_{(s,a,\tilde{s},\bar{s}) \sim \mu_h} \ell(f'(\phi(s)), g(\tilde{s}), g(\bar{s}), a)$ for $f' \in$

$\mathcal{F}'$ and similarly define $\hat{\ell}_h'^{\mathbf{x}}(\phi, f', g) = \hat{\ell}_h^{\mathbf{x}}(\phi, \texttt{softmax}(f'), g)$. Notice that $\ell(\cdot, \alpha, \beta, a)$ is $2K$-lipschitz, $\ell(\boldsymbol{v}, \cdot, \beta, a)$ is $K$-lipschitz and $\ell(\boldsymbol{v}, \alpha, \cdot, a)$ is 1-lipschitz, Using Theorem 8(i) from Maurer et al. (2016), we get that

$$\mathop{\mathbb{E}}_{\mu \sim \eta} \mathop{\mathbb{E}}_{\mathbf{x} \sim \mu^n} \sup_{f \in \mathcal{F}} \sup_{g \in \mathcal{G}} \left[ \hat{\ell}_h^{\mathbf{x}}(\phi, f, g) - \ell_h^\mu(\phi, f, g) \right]$$

$$= \mathop{\mathbb{E}}_{\mu \sim \eta} \mathop{\mathbb{E}}_{\mathbf{x} \sim \mu^n} \sup_{f' \in \mathcal{F}'} \sup_{g \in \mathcal{G}} \left[ \hat{\ell}_h'^{\mathbf{x}}(\phi, f', g) - \ell_h'^\mu(\phi, f', g) \right]$$

$$\leq \frac{\sqrt{2\pi}\, \mathbb{E}_{\mathbf{x}}\, G(\ell(\mathcal{F}'(\phi(\mathbf{s}_h)), \mathcal{G}(\bar{\mathbf{s}}_h), \mathcal{G}(\tilde{\mathbf{s}}_h), \mathbf{a}))}{n}$$

$$\leq \frac{2\sqrt{2\pi}KG(\mathcal{F}'(\phi(\mathbf{s}_h)), \mathbf{a})}{n} + \mathop{\mathbb{E}}_{\mu \sim \eta} \mathop{\mathbb{E}}_{\mathbf{x} \sim \mu^n} \left[ \frac{\sqrt{2\pi}KG(\mathcal{G}(\bar{\mathbf{s}}_h))}{n} + \frac{\sqrt{2\pi}G(\mathcal{G}(\tilde{\mathbf{s}}_h))}{n} \right]$$

$$\leq c\frac{RK\sqrt{K}}{\sqrt{n}} + c' \mathop{\mathbb{E}}_{\mu \sim \eta} \mathop{\mathbb{E}}_{\mathbf{x} \sim \mu^n} \left[ \frac{KG(\mathcal{G}(\bar{\mathbf{s}}_h))}{n} + \frac{G(\mathcal{G}(\tilde{\mathbf{s}}_h))}{n} \right] \leq \epsilon_{gen,h}(\mathcal{F}, \mathcal{G})$$

where we used lipschitzness and Slepian's lemma for second inequality and a similar computation to Lemma A.1 for the third. $\square$

*Proof of Lemma B.4.*

$$\bar{L}_h(\phi) - \mathop{\mathbb{E}}_{\mathbf{x} \sim \rho_h} \hat{m}_{\mathbf{x}}(\phi) = \mathop{\mathbb{E}}_{\mu \sim \eta} \mathop{\mathbb{E}}_{\mathbf{x} \sim \mu^n} \bar{m}_{\mu,\mathbf{x}}(\phi) - \mathop{\mathbb{E}}_{\mu \sim \eta} \mathop{\mathbb{E}}_{\mathbf{x} \sim \mu^n} \hat{m}_{\mathbf{x}}(\phi)$$

$$= \mathop{\mathbb{E}}_{\mu \sim \eta} \mathop{\mathbb{E}}_{\mathbf{x} \sim \mu^n} \max_{g \in \mathcal{G}} \ell_h^\mu(\phi, \hat{f}_{\mathbf{x}}^\phi, g) - \max_{g \in \mathcal{G}} \hat{\ell}_h^{\mathbf{x}}(\phi, \hat{f}_{\mathbf{x}}^\phi, g)$$

$$\leq \mathop{\mathbb{E}}_{\mu \sim \eta} \mathop{\mathbb{E}}_{\mathbf{x} \sim \mu^n} \max_{g \in \mathcal{G}} [\ell_h^\mu(\phi, \hat{f}_{\mathbf{x}}^\phi, g) - \hat{\ell}_h^{\mathbf{x}}(\phi, \hat{f}_{\mathbf{x}}^\phi, g)]$$

$$\leq \mathop{\mathbb{E}}_{\mu\sim\eta} \mathop{\mathbb{E}}_{\mathbf{x}\sim\mu^n} \max_{f\in\mathcal{F}} \max_{g\in\mathcal{G}}[\ell_h^\mu(\phi,f,g) - \hat{\ell}_h^{\mathbf{x}}(\phi,f,g)]$$

$$\leq \epsilon_{gen,h}(\mathcal{F},\mathcal{G})$$

where we use the definition of $\bar{L}_h$, obviousness for the first inequality and Lemma B.3 for the last. $\qquad\square$

*Proof of Lemma B.5.* We wil be using Slepian's lemma

**Lemma B.7** (Slepian's lemma). *Let $\{X\}_{s\in S}$ and $\{Y\}_{s\in S}$ be zero mean Gaussian processes such that*

$$\mathbb{E}(X_s - X_t)^2 \leq \mathbb{E}(Y_s - Y_t)^2, \forall s,t \in S$$

*Then*

$$\mathbb{E}\sup_{s\in S} X_s \leq \mathbb{E}\sup_{s\in S} Y_s$$

Using Theorem 8(ii) from Maurer et al. (2016), we get that

$$\sup_{\phi\in\Phi}\left[\mathop{\mathbb{E}}_{\mathbf{x}\sim\rho_h} \hat{m}_{\mathbf{x}}(\phi) - \frac{1}{T}\sum_i \hat{m}_{\mathbf{x}^{(i)}}(\phi)\right] \leq \frac{\sqrt{2\pi}}{T}G(S) + \sqrt{\frac{9\ln(2/\delta)}{2T}} \tag{12}$$

where $S = \{(\hat{m}(\phi)_{\mathbf{x}_1},\ldots,\hat{m}(\phi)_{\mathbf{x}_T}) : \phi \in \Phi\}$. We bound the Gaussian average of $S$ using Slepian's lemma. Define two Gaussian processes indexed by $\Phi$ as

$$X_\phi = \sum_i \gamma_i \hat{m}(\phi)_{\mathbf{x}^{(i)}} \text{ and } Y_\phi = \frac{2K}{\sqrt{n}}\sum_i \gamma_{ijk}\phi(s_j^i)_k$$

For $\mathbf{x} = \{(s_j, a_j, \tilde{s}_j, \bar{s}_j)\}$, consider 2 representations $\phi$ and $\phi'$,

$$(\hat{m}(\phi)_{\mathbf{x}} - \hat{m}(\phi')_{\mathbf{x}})^2 = (\min_{f\in\mathcal{F}}\max_{g\in\mathcal{G}}\hat{\ell}_h^{\mathbf{x}}(\phi,f,g) - \min_{f\in\mathcal{F}}\max_{g\in\mathcal{G}}\hat{\ell}_h^{\mathbf{x}}(\phi',f,g))^2$$

$$\leq (\sup_{f\in\mathcal{F},g\in\mathcal{G}}|\hat{\ell}_h^{\mathbf{x}}(\phi,f,g) - \hat{\ell}_h^{\mathbf{x}}(\phi',f,g)|)^2$$

$$= \left(\sup_{f\in\mathcal{F},g\in\mathcal{G}}\left|\frac{1}{n}\sum_j[K\pi^{\phi,f}(a_j|s_j)g(\tilde{s}_j) - K\pi^{\phi',f}(a_j|s_j)g(\tilde{s}_j)]\right|\right)^2$$

$$= K^2\left(\sup_{f\in\mathcal{F},g\in\mathcal{G}}\left|\frac{1}{n}\sum_j(f(\phi(s_j))_{a_j} - f(\phi'(s_j))_{a_j})g(\tilde{s}_j)\right|\right)^2$$

$$\leq \frac{K^2}{n}\sup_{f\in\mathcal{F}}\sum_j\left(f(\phi(s_j))_{a_j} - f(\phi'(s_j))_{a_j}\right)^2$$

$$\leq \frac{4K^2}{n}\sum_j|\phi(s_j) - \phi'(s_j)|^2 = \frac{4K^2}{n}\sum_{j,k}(\phi(s_j)_k - \phi'(s_j)_k)^2$$

where we prove the first inequality later, second inequality comes from $g$ being upper bounded by 1 and by Cauchy-Schwartz inequality, third inequality comes from the 2-lipschitzness of $f$.

$$\mathbb{E}(X_\phi - X_{\phi'}) = \sum_i(\hat{m}(\phi)_{\mathbf{x}^{(i)}} - \hat{m}(\phi')_{\mathbf{x}^{(i)}})^2$$

$$\leq \frac{4K^2}{n}\sum_{i,j,k}(\phi(s_j^i)_k - \phi'(s_j^i)_k)^2 = \mathbb{E}(Y_\phi - Y_{\phi'})^2$$

Thus by Slepian's lemma, we get

$$G(S) = \mathbb{E}\sup_{\phi\in\Phi} X_\phi \leq \mathbb{E}\sup_{\phi\in\Phi} Y_\phi = \frac{2K}{\sqrt{n}}G(\Phi(\{s_j^i\}))$$

Plugging this into Equation 12 completes the proof. To prove the first inequality above, notice that

$$\min_{f \in \mathcal{F}} \max_{g \in \mathcal{G}} \hat{\ell}_h^{\mathbf{x}}(\phi, f, g) - \min_{f \in \mathcal{F}} \max_{g \in \mathcal{G}} \hat{\ell}_h^{\mathbf{x}}(\phi', f, g) = \hat{\ell}_h^{\mathbf{x}}(\phi, f, g) - \hat{\ell}_h^{\mathbf{x}}(\phi', f', g')$$
$$\leq \hat{\ell}_h^{\mathbf{x}}(\phi, f', g'') - \hat{\ell}_h^{\mathbf{x}}(\phi', f', g')$$
$$\leq \hat{\ell}_h^{\mathbf{x}}(\phi, f', g'') - \hat{\ell}_h^{\mathbf{x}}(\phi', f', g'')$$
$$\leq \sup_{f \in \mathcal{F}, g \in \mathcal{G}} |\hat{\ell}_h^{\mathbf{x}}(\phi, f, g) - \hat{\ell}_h^{\mathbf{x}}(\phi', f, g)|$$

By symmetry, we also get that $\min_{f \in \mathcal{F}} \max_{g \in \mathcal{G}} \hat{\ell}_h^{\mathbf{x}}(\phi, f, g) - \min_{f \in \mathcal{F}} \max_{g \in \mathcal{G}} \hat{\ell}_h^{\mathbf{x}}(\phi', f, g) \leq \sup_{f \in \mathcal{F}, g \in \mathcal{G}} |\hat{\ell}_h^{\mathbf{x}}(\phi, f, g) - \hat{\ell}_h^{\mathbf{x}}(\phi', f, g)|$. $\qquad\square$

*Proof of Lemma B.6.* Let $\bar{g} = \arg\max_{g \in \mathcal{G}} \left( \mathbb{E}_{s \sim \nu_h^{\pi}} \Gamma_h^{\boldsymbol{\pi}} g(s) - \mathbb{E}_{s \sim \nu_h^*} \Gamma_h^{\boldsymbol{\pi}} g(s) \right)$ and $g' = \arg\min_{g \in \mathcal{G}} |g - \Gamma_h^{\boldsymbol{\pi}} \bar{g}|_{(\nu_h^{\pi} + \nu_h^*)/2}$.

$$\max_{g \in \mathcal{G}} \left( \mathbb{E}_{s \sim \nu_h^{\pi}} \Gamma_h^{\boldsymbol{\pi}} g(s) - \mathbb{E}_{s \sim \nu_h^*} \Gamma_h^{\boldsymbol{\pi}} g(s) \right) = \mathbb{E}_{s \sim \nu_h^{\pi}} \left[ \Gamma_h^{\boldsymbol{\pi}} \bar{g}(s) - \mathbb{E}_{s \sim \nu_h^*} \Gamma_h^{\boldsymbol{\pi}} \bar{g}(s) \right]$$
$$\leq | \mathbb{E}_{s \sim \nu_h^{\pi}} g'(s) - \mathbb{E}_{s \sim \nu_h^*} g'(s)| + | \mathbb{E}_{s \sim \nu_h^{\pi}} [g'(s) - \Gamma_h^{\boldsymbol{\pi}} \bar{g}(s)]| + | \mathbb{E}_{s \sim \nu_h^*} [g'(s) - \Gamma_h^{\boldsymbol{\pi}} \bar{g}(s)]|$$
$$\leq \max_{g \in \mathcal{G}} | \mathbb{E}_{s \sim \nu_h^{\pi}} g(s) - \mathbb{E}_{s \sim \nu_h^*} g(s)| + 2 \mathbb{E}_{s \sim (\nu_h^{\pi} + \nu_h^*)/2} [|g'(s) - \Gamma_h^{\boldsymbol{\pi}} \bar{g}(s)|]|$$
$$\leq \Delta_h + 2\epsilon_{be}^{\pi}$$

$\qquad\square$

# C    DATA SET COLLECTION DETAILS

## C.1    DATASET FROM TRAJECTORIES

Given $n$ expert trajectories for a task $\mu$, for each trajectory $\tau = (s_1, a_1, \ldots, s_H, a_H)$ we can sample an $h \sim \mathcal{U}([H])$ and select the pair $(s_h, a_h)$ from that trajectory[7]. This gives us $n$ i.i.d. pairs $\{(s_j, a_j)\}_{j=1}^n$ for the task $\mu$. We collect this for $T$ tasks and get datasets $\mathbf{x}^{(1)}, \ldots, \mathbf{x}^{(T)}$.

## C.2    DATASET FROM TRAJECTORIES AND INTERACTION

Given $2n$ expert trajectories for a task $\mu$, we use first $n$ trajectories to get independent samples from the distributions $\nu_{1,\mu}^*, \ldots, \nu_{H,\mu}^*$ respectively for the $\bar{s}$ states in the dataset. Using the next $n$ trajectories, we get samples from $\nu_{0,\mu}^*, \ldots, \nu_{H-1,\mu}^*$ for the $s$ states in the dataset, and for each such state we uniformly sample an action $a$ from $\mathcal{A}$ and then get a state $\tilde{s}$ from $P_{s,a}$ by resetting the environment to $s$ and playing action $a$. We collect this for $T$ tasks and get datasets $\mathbf{X}^{(i)} = \{\mathbf{x}_1^{(i)}, \ldots, \mathbf{x}_H^{(i)}\}$ for every $i \in [T]$, where each dataset $\mathbf{x}_h^{(i)}$ a set of $n$ tuples obtained level $h$. Rearranging, we can construct the datasets $\mathbf{X}_h = \{\mathbf{x}_h^{(1)}, \ldots, \mathbf{x}_h^{(T)}\}$.

# D    EXPERIMENT DETAILS

For the policy optimization experiments, we use 4 random seeds to evaluate our algorithm. We show the results for 1 test environment as the results for other test environments are also showing the algorithm works but the magnitude of reward might be different, so we do not average the numbers over different test environments.

---

[7]In practice one can use all pairs from all trajectories, even though the samples are not strictly i.i.d.

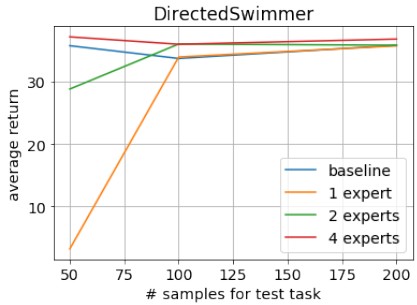

Figure 3: The total rewards by different algorithms in DirectedSwimmer.

**Experiment Setup** We first describe the construction of the NoisyCombinationLock environment. The state space is $\mathbb{R}^{40}$. Each state $s$ is in the form of $[s_{\text{real}}, s_{\text{noise}}]$, while $s_{\text{real}} \in \mathbb{R}^{20}$ is either a one-hot vector or a zero vector, and $s_{\text{noise}} \in \mathbb{R}^{20}$ is sampled from $\mathcal{N}(\mathbf{0}, \mathbf{I})$. The action space is discrete and has size 2. For each MDP, we have a sequence of actions $\mathbf{a}^* \in [2]^{20}$. This is the sequence of optimal actions. We use different $\mathbf{a}^*$ to define different environments. The transition model is that: If $s_{\text{real}} = e_i$ for some $i$ and the action is $\mathbf{a}_i^*$, then $s'_{\text{real}} = e_{i+1}$ and we'll get reward 1. Otherwise $s'_{\text{real}}$ will be all zero and the reward is 0. $s_{\text{noise}}$ will always be sampled from the Gaussian distribution. Note that once $s_{\text{real}}$ is all zero, it will not change and the reward will always be 0. The maximum horiozn is set to 20 and therefore, the optimal policy has return 20. The initial $s_{\text{real}}$ is always $e_1$.

The representation has dimension of 10. We limit the function $\phi$ to be a linear mapping from $\mathbb{R}^{40}$ to $\mathbb{R}^{10}$. Although the dimension of representation is smaller than the number of states, there still exists a linear mapping from states to representation such that we can find a linear optimal policy. For each expert, we collect 200 state-action pairs to train the representation $\phi$. The trajectories are generated by the optimal policy.

When training the policy using an RL algorithm, to reduce the impact of initialization, the last full connected layer is initialized to 0. We use the PPO (Schulman et al., 2017) algorithm to train our policy with code from Dhariwal et al. (2017).

**DirectedSwimmer** A DirectedSwimmer environment is the same as Swimmer in OpenAI Gym (Brockman et al., 2016), except the following: the reward function is parametrized by a direction $d$ with $\|d\| = 1$, and is defined as the traveled distance along the direction $d$. For each task, we sample a random direction. The state space is still $\mathbb{R}^8$. The original action space in Swimmer is $\mathbb{R}^2$, and we discretize the action space, such that each entry can be only one of $\{-1, -0.5, 0, 0.5, 1\}$. We also reduce the maximum horizon from 1000 to 100. We trained the experts for 1 million steps by PPO to make sure it converges.

The function $\phi$ we use has two fully connected layers and two ReLU layers. The number of hidden units is 100, so is the dimension of representation. We also include the total rewards that each algorithm can get in Figure 3. Note even though the baseline has a high validation loss, its performance can be quite good. This does not indicate a failure of representation learning, but it shows that lower logistic loss does not always imply higher reward.

**Optimization** All optimization, including training $\phi, \pi$ and behavior cloning baseline, is done by Adam (Kingma & Ba, 2014) with learning rate 0.001 until it converges. To solve equation 8, we build a joint loss over $\phi$ and all $f$'s in each task,

$$\mathcal{L}(\phi, f_1, \ldots, f_T) = \frac{1}{nT} \sum_{t=1}^{T} \sum_{j=1}^{n} -\log(\pi^{\phi, f_t}(s_j^t)_{a_j^t}). \tag{13}$$

Then we minimize $\mathcal{L}(\phi, f_1, \ldots, f_T)$ and obtain the optimal $\phi$.

