# OpenReview forum: "Provable Representation Learning for Imitation Learning via Bi-level Optimization"
_ICLR.cc/2020/Conference — Reject_

### Official Review · AnonReviewer3 · 2019-10-20
**Official Blind Review #3**

**Rating:** 3

**Review:**

(I bid on the paper thinking that bi-level optimisation would play a major role in the paper. Unfortunately, it does not, so my expertise in bi-level optimisation is not much use, I am afraid.)

The authors study applying policies learned on one task to another task, while considering practical finite-sample limitations. They call this the "representatition learning for imitation learning". Unfortunately:

The make extensive assumptions, summarised on page 3, but not formalised as Assumptions 1, 2, 3 anywhere, as far as I can tell. They assume:
-- concentration of the loss, "which guarantees within-task sample efficiency" (but does not seem easy to support in practice? actually, one may observe samples from a stochastic process, rather than iid samples with any concentration what-so-ever?),
-- that the "loss" they use with is somehow close to the optimum of the expected value function J (for which I again see no justification, empirical or otherwise).

Then they reuse results of Maurer et al (2016) and extend them to a case where the actions are not observable. The results are plausible, given the assumptions. Given the assumptions, they also do not seem to be particularly relevant to the practice of RL?

The empirical results involve only benchmarks of the authors own coinage, and hence are hard to evaluate. It seems plausible, again, however, that the approach may work in some cases.



**Experience Assessment:**

I do not know much about this area.

**Review Assessment: Checking Correctness Of Derivations And Theory:**

I assessed the sensibility of the derivations and theory.

**Review Assessment: Checking Correctness Of Experiments:**

I assessed the sensibility of the experiments.

**Review Assessment: Thoroughness In Paper Reading:**

I made a quick assessment of this paper.

---

> ### Author Response · Authors · 2019-11-12
> **Response**
>
> Thank you for review. The three properties listed on page 3 are just high-level descriptions of the sufficient conditions that enable us to show reduced sample complexity for representation learning. We *do not assume* these properties, but we *prove* that they hold for both the standard settings of behavioral cloning and observations only. Abstracting the proof into these three properties gives us a general recipe to potentially prove such a theorem for other settings as well. We will add clarifications on these properties in the revision to avoid confusions.
>
> The precise assumptions for Theorem 4.1 are stated in Section 4 and assumptions for Theorem 5.1 are stated in Section 5.

---

### Official Review · AnonReviewer1 · 2019-10-23
**Official Blind Review #1**

**Rating:** 6

**Review:**

This paper theoretically explores reinforcement/imitation learning via representation learning. The key theoretical question being investigated is the relationship between representation learning in a multi-task/meta learning setup and its dependence to the sample/task complexity. The paper sets up the problem in bilevel optimization framework, where the inner optimization learns/optimizes task specific losses, while the outer optimization learns the representation used in the inner level tasks. The main takeaway from the two theorems (which are the core contributions of this paper) are that when the number of tasks is higher than the number of samples, representation learning can reduce the sample complexity. The paper explores two scenarios in imitation learning, namely behavioral cloning and when only the states of the experts are available (and not their actions). Some experiments are provided to empirically validate the theory.

Pros:
1. The paper presents a theoretical investigation into multi-task/meta learning for RL via learning representations. While, the main theoretical contributions are perhaps marginal with regard to prior work, the problem setting (RL) seems novel and the two theorems in this context are interesting.
2. The paper is well-written and appears to be very rigorous. I did not check for the correctness of all technical parts. There are several "abuse of notations" in the main text, which sometimes impact the otherwise smooth read of the technical parts.
3. A good and concise review of RL concepts is provided.

Cons:
1. The paper lacks a good literature review to place this work in the right context. For example, while the paper refers to the works of Maurer 2016 and Sun et al. 2019 at several places, it is not formally and clearly mentioned anywhere what are the similarities to these prior works and what are the new contributions. For example, Maurer 2016 proposes a multi-task learning setup using representation learning, and most of Theorem 4.1 in this paper is taken from the results in that paper. While, the current paper uses bilevel optimization setting in an RL context, it is not clear to me if this (bilevel + RL) setting has any significant bearing against the theoretical results furnished by Maurer 2016. For example, the theorems in this paper (as far as I see) show that the bounds are scaled by a constant defined by H^2, the trajectory length? If there is something beyond this, then the paper needs to explicitly point it out. The same comment goes with the results against Sun et al. 2019 in Theorem 5.1.

2. Given that the main goal of this paper is to connect sample complexity with representation learning, it is important the paper provide a theorem stating this precisely. Theorems 4.1 and 5.1 provide a general bound, and the sample complexity is being described in the explanations of this theorem, which is very informal. Also, against what is claimed in the abstract, it appears that representation learning helps reduce sample complexity only when the number of tasks are larger, which perhaps needs to be explicitly mentioned. Also, note that there is a bearing of the bound on the trajectory length H (Theorem 4.1, and 5.1). Shouldn't this factor be also accounted for when explaining the sample complexity?

3.  There has been several recent works on model-agnostic meta-learning (that also uses bilevel optimization and implicit gradients), however, older works on meta-learning (only for imitation learning) have been cited. The paper should include more recent works in this area and contrast against their theoretical findings.

Apart from these, below are some minor comments that could help improve the reading of this paper:
a. Theorem 3.1 is not really a theorem, since it is very informal. Also, fix the Theorem numbers.

b. Bullet 1. after Theorem 3.1, \ell^x(\pi) concentrates to \ell^x(\pu*). Also, the mention about sample complexity here should be backed with some reference/citation.

c. Assumption 4.2: The notation \pi_\mu(s)_{\pi*(\mu(s)) is unclear, shouldn't the subscript contain an argmax over the actions for \pi*?

d. Theorem 4.1 and 5.1, what does it mean by "probability 1-\delta over the choice of the dataset X" ? Also, \mu^n seems undefined.

e. The first two terms in Theorem 4.1 are claimed standard, provide the citations?

f. Theorem 4.1, perhaps use some other notation for c, which is defined as the cost/reward in the RL setting.

Overall, the paper has some interesting theoretical results, and is mathematically rigorous, however lacks a clear distinction from prior and more recent works in this area.


**Experience Assessment:**

I have read many papers in this area.

**Review Assessment: Checking Correctness Of Derivations And Theory:**

I assessed the sensibility of the derivations and theory.

**Review Assessment: Checking Correctness Of Experiments:**

I assessed the sensibility of the experiments.

**Review Assessment: Thoroughness In Paper Reading:**

I read the paper at least twice and used my best judgement in assessing the paper.

---

> ### Author Response · Authors · 2019-11-12
> **Response**
>
> We thank you for the careful and positive review. We will modify the paper according to your comments. Please find our responses to your comments below.
>
> 1. “The paper lacks a good literature review to place this work in the right context.” We will add more discussions on the literature. Our work bridges the multitask representation learning literature for supervised learning (Maurer et al., 2016) and single task imitation learning methods (Ross and Bagnell, 2010; Sun et al., 2019). The additional factor of H^2 is incurred while connecting the imitation loss function to the total cost in the MDP; this factor of H^2 is common in imitation learning and occurs both in Ross and Bagnell, 2010 and Sun et al., 2019. The bi-level framework is an abstraction of Maurer et al. that lets us go beyond supervised learning losses and can potentially be used for other imitation and reinforcement learning settings. We will make these points clearer in the revision.
>
> 2. It is straightforward to derive PAC-style sample complexity bounds using the existing bounds in Theorem 4.1 and Theorem 5.1. We will add such a bound in our next version soon. Again the H factor comes from the error propagation in imitation learning.
>
> 3. We will add more discussions on gradient based meta-learning and bi-level optimization. One key difference from previous theoretical analyses is that they either deal with computational complexity (which we do not), or show sample complexity/regret guarantees for *convex* losses, whereas our analysis can deal with any function class.
>
> We will fix other minor issues accordingly. Thanks for carefully reading the paper and pointing them out!

---

### Official Review · AnonReviewer4 · 2019-11-07
**Official Blind Review #4**

**Rating:** 3

**Review:**

Overview:

The paper tackles the representation learning problem where the aim is to learn a generic representation that is useful for a variety of downstream tasks. A two-level optimization framework is proposed: an inner optimization over the specific problem-at-hand, and an outer optimization over other similar problems. The problem is studied in two settings of the imitation learning framework with the additional aim of providing mathematical guarantees in terms of sample efficiency on new tasks. An extensive theoretical analysis is performed, and some preliminary empirical results are presented.

Decision:

In its current form, the paper should be rejected because (1) the empirical analysis is incomplete – the baseline isn't very appropriate, the results are not conclusive, details are scattered or not included, (2) the literature survey does not connect the proposed approach with existing approaches, and does not convince the reader why all the existing approaches have not been compared against empirically, (3) the paper is generally unpolished and needs more work before being considered for acceptance.

Details:

The paper makes both theoretical and empirical claims. I did not have the time to thoroughly verify the theoretical claims and took them at face value. I consider the theoretical guarantees associated with the proposed approach a welcome and valuable contribution to this field that has recently been relying primarily on limited empirical work to assess any method.

The empirical results presented in the paper do not sufficiently support the claims of sample efficiency. One of the main issues with the empirical analysis is the choice of the baseline, which learns a policy from scratch. This does not help make conclusions about the sample efficiency of the proposed method on new tasks. A better baseline would be one that learns some representation from the T previous tasks, which would help infer if the proposed method to learn representations is actually more sample efficient on new tasks or not. There is also no comparison with existing approaches that are mentioned in the Related Work section. If those aren’t appropriate baselines for this problem, a small explanation of the reasons why would help readers understand why they haven’t been compared against. Additionally, an analysis of statistical significance of the results is missing and would significantly help in gauging the efficacy of the proposed approach.

The paper notes that these are some preliminary experiments. The completion of the empirical analysis would definitely make a stronger case for this paper to be accepted.

Minor comments to improve the paper:

- Error bars in the plot, specification of number of runs, and other such experimental details would be very helpful in interpreting the results.
- It would help a reader if the paper was more self-contained, e.g., if terms like supp(\eta), \bar{s}, \tilde{s} are defined more clearly.
- It would also help to say what the proofs intuitively mean, e.g., for a new task drawn from this particular distribution of tasks, the agent would achieve close-to-X performance within Y samples – something along those lines.
- There are some typos, e.g., 'possibility'->'possibly' on page 1, missing $H$ in specification of MDPs on page 2, 'exiting'->'exciting' on page 8, some latex symbols in Appendix D, etc.
- The bibliography has a lot of issues – some references are incorrectly parsed (e.g., Yan Duan, Marcin Andrychowicz, Bradly Stadie, Jonathan Ho, Jonas Schneider, Ilya Sutskever, Pieter Abbeel, and Wojciech Zaremba. One-shot imitation learning. 03 2017), others are inconsistent (e.g., "In NIPS" and "In Advances in Neural…”; the arXiv ones).



**Experience Assessment:**

I have read many papers in this area.

**Review Assessment: Checking Correctness Of Derivations And Theory:**

I did not assess the derivations or theory.

**Review Assessment: Checking Correctness Of Experiments:**

I carefully checked the experiments.

**Review Assessment: Thoroughness In Paper Reading:**

I read the paper at least twice and used my best judgement in assessing the paper.

---

> ### Author Response · Authors · 2019-11-12
> **Response**
>
> We thank you for the detailed review. First, we would like to emphasize that the main aim of this paper is to demonstrate the statistical advantage of representation learning for imitation learning, in a mathematically rigorous way. The experiments in our paper, like many machine learning theory papers, are meant as proof-of-concepts and mainly verify the theoretical results.
>
> Please find our responses to your detailed comments below. We will add more clarifications to avoid confusions in our next version soon.
>
> Comparing with other baselines:
> We would like to clarify that we do not propose any new algorithm. The bi-level optimization framework is introduced for problem formulation and ease of analysis. The algorithms we run are all natural extensions of representation learning methods for supervised learning, with a few tweaks. We do not claim that our algorithm is more sample efficient than existing meta learning algorithms or that it beats them.
>
> “A better baseline would be one that learns some representation from the T previous tasks, which would help infer if the proposed method to learn representations is actually more sample efficient on new tasks or not.”: In fact, the algorithm we test is precisely of this form. The algorithms mentioned in related work section do not learn a fixed representation and hence we do not test these methods. We will clarify this point and add a more detailed comparison to some previous works in our revision soon.
>
> Minor comments:
> We will add error bars to our plots, add some more intuition for results and fix other formatting issues and typos. Thanks for pointing them out!

---

> > ### Comment · AnonReviewer4 · 2019-11-13
> > **Some more comments/questions based on the authors' response**
> >
> > Thank you for taking the time to respond to my previous comments. Based on your responses, I had some more comments/questions:
> >
> > "First, we would like to emphasize that the main aim of this paper is to demonstrate the statistical advantage of representation learning for imitation learning, in a mathematically rigorous way." and
> > "The algorithms we run are all natural extensions of representation learning methods for supervised learning, with a few tweaks."
> > I see. This wasn't clear from the wording used in the paper, e.g., in the Introduction, it is mentioned that "*Furthermore*, the framework allows us to do a rigorous analysis to show provable benefits of representation learning for imitation learning", which indicates that this is an additional contribution.
> > Given the above statements, I think the wording should be re-phrased in order to draw the line between existing work and this work more clearly.
> >
> > You also mention that you do not propose any new algorithm. Even if that is the case, it would greatly help a reader if the algorithm used to conduct the experiments were explicitly specified in pseudo-code form or something. Additionally, the baselines used in both tasks should also be explicitly specified to help the reader gauge the benefits of a new/different approach. Because of this (and the points mentioned previously), as of now, it is a little hard to interpret the experimental section and draw concrete conclusions from it, despite the rigorous theoretical work.
> >
> > An additional question based on the experiments: is there a particular reason why data from more experts was not used in DirectedSwimmer? There is some evidence that the baseline performs better as compared to learning representations using data from <=8 experts in both the domains, but a larger number of experts seems to help in the second domain, NoisyCombinationLock. Did you see a similar trend in DirectedSwimmer as well?

---

> > > ### Author Response · Authors · 2019-11-15
> > > **Response**
> > >
> > > Thanks for responding! We have uploaded a revision and re-phrased the problematic wording in the introduction.
> > >
> > > We added more description about the representation algorithm we test and the baseline that we compare against in the experiments section. Furthermore, we added implementation details in the appendix.
> > >
> > > Your suspicion is correct and indeed we see further improvement in DirectedSwimmer by using 16 experts (we updated the plot to include this as well). Note, however, that the main point of this experiment is to show that representation helps learn a good policy *much faster* (i.e. with fewer samples) than learning a policy from scratch (i.e. the baseline). This is true even for 8 experts in the previous plot (before 20K steps). However, it is inevitable for the baseline to eventually perform the best with many more samples, since it is allowed to learn an optimal representation from scratch. We updated the plot in the revision by zooming into the initial stages so that the benefit of representation learning can be seen more clearly in the few samples regime.

---

### Author Response · Authors · 2019-11-15
**Revision uploaded**

We thank all the reviewers for providing useful feedback and comments! We made the following main changes in our revision

- Clarified our precise contribution in the introduction (as pointed out by reviewer #4) and made a more detailed comparison with recents works in the related work section, based on feedback from reviewer #1. The key difference is that *previous work* either showed guarantees for multi-task *supervised learning* or *single task* imitation learning or showed guarantees for gradient based meta-learning methods *only for convex losses*. Our analysis can show guarantees for multi-task imitation learning methods for arbitrary representation function classes that can make the loss non-convex.

- Added PAC style sample complexity bounds in Corollaries 5.1 and 6.1 for the simple case of finite representation function class and gave some more intuition about what the theorem statements mean. Hope this clarifies the points raised by both reviewers #1 and #4.

- Added more details about the experimental setup, emphasized our algorithm and the baseline method, and included error bars in the plots, as requested by reviewer #4.

- Rephrased the discussion about the 3 properties right after informal theorem 3.1. In particular, these properties are not assumptions that we make, but we prove them for our two settings. Hopefully the revision clarifies the confusion that reviewer #2 had.

- Fixed typos, notations and formatting issues that were pointed out by reviewers #1 an #4

---

### Decision · Program_Chairs · 2019-12-19

**Decision:**

Reject

**Comment:**

This paper proposes a methodology for learning a representation given multiple demonstrations, by optimizing the representation as well as the learned policy parameters. The paper includes some theoretical results showing that this is a sensible thing to do, and an empirical evaluation.

Post-discussion, the reviewers (and me!) agreed that this is an interesting approach that has a lot of promise. But there was still concern about he empirical evaluation and the writing. Hence I am recommending rejection.